# Automated Road Defect and Anomaly Detection for Traffic Safety: A Systematic Review

**DOI:** 10.3390/s23125656

**Published:** 2023-06-16

**Authors:** Munish Rathee, Boris Bačić, Maryam Doborjeh

**Affiliations:** 1School of Engineering, Computer and Mathematical Sciences, Auckland University of Technology, Auckland 1142, New Zealand; mgholami@aut.ac.nz; 2Knowledge Engineering and Discovery Research Innovation, Auckland University of Technology, Auckland 1142, New Zealand

**Keywords:** on-road anomaly detection, structural damage detection, motorist safety, computer vision, machine learning, deep learning, transfer learning, ARDAD

## Abstract

Recently, there has been a substantial increase in the development of sensor technology. As enabling factors, computer vision (CV) combined with sensor technology have made progress in applications intended to mitigate high rates of fatalities and the costs of traffic-related injuries. Although past surveys and applications of CV have focused on subareas of road hazards, there is yet to be one comprehensive and evidence-based systematic review that investigates CV applications for Automated Road Defect and Anomaly Detection (ARDAD). To present ARDAD’s state-of-the-art, this systematic review is focused on determining the research gaps, challenges, and future implications from selected papers (N = 116) between 2000 and 2023, relying primarily on Scopus and Litmaps services. The survey presents a selection of artefacts, including the most popular open-access datasets (D = 18), research and technology trends that with reported performance can help accelerate the application of rapidly advancing sensor technology in ARDAD and CV. The produced survey artefacts can assist the scientific community in further improving traffic conditions and safety.

## 1. Introduction

Traffic accidents caused by road surface defects or unwanted objects lead to deaths, injuries and billions of dollars in property damage [1,2,3,4]. According to Justo-Silva and Ferreira [4], over 1.25 million lives are lost, and 20 to 50 million people are injured annually in traffic accidents worldwide. Moreover, highway accidents are predicted to be the fifth-highest cause of mortality by 2030. A 2019 survey based on approximately 166 countries by Chen et al. [5] estimated that road injuries would cost the world economy USD 1.8 trillion from 2015 to 2030, equivalent to a 0.12% annual tax on the global gross domestic product. Mohammed et al. [6] found that road accidents are now one of the top three causes of predicted deaths, posing a global threat to lives and economies. Among the multiple causes of crashes reported by the American Association of State and Highway Transportation Officials (AASHTO) [7], roadway factors such as road defects and anomalies account for approximately 34% [4].

The scientific community’s aim to help reduce road accidents by detecting surface defects and predicting anomalies has existed since the advent of high-speed roads. A positive shift in momentum started with the advancements of sensor technology and the application of computer vision (CV) combined with soft-computing approaches such as machine learning (ML) and deep learning (DL) for adaptive automated road defect and anomaly detection (ARDAD) systems. As a consumer-grade example, modern mobile phones are equipped with features such as inertial sensors, high-speed video, and other sensors such as light detection and ranging (LiDAR).

The first contribution of this systematic review is the discovery of an upward trend in surveillance automation since 2000, with a correlation between the scientific community’s growing interest and technological advancement.

ARDAD systems can significantly ease the day-to-day maintenance process and reduce the loss of life and costs associated with traffic-related injuries [3]. However, despite the growing number of publications on ARDAD systems since 2020, most surveys focus on one or two of many problem domains, such as (a) road surface cracks [8,9,10], (b) road surface defects [10,11,12], (c) structural damage [13,14], or (d) anomaly detection [15,16,17,18,19].

As a second contribution, our systematic review uniquely combines all ARDAD methods and focuses on traffic safety impacted by various on-road hazards (Figure 1). Overview of automated anomaly/defect detection process. This approach distinguishes our review from others in the field and provides a comprehensive analysis of the current state-of-the-art ARDAD systems, making it a valuable resource for researchers and professionals working in the field of traffic safety.

### 1.1. Background

Road surfaces are constructed using different materials, which degrade over time due to wear, environmental effects, or external factors. Figure 2 provides a generally established unifying process of automated anomaly/defect detection. To ensure safety and maintain infrastructural integrity, various types of structural damage (Figure 3) must be regularly monitored and addressed to determine the underlying causes. Structural damage caused by poor construction techniques or external factors may take the form of potholes, cracks (due to thermal action), debonding, stripping, ravelling, bleeding, shrinkage of road layers, and swelling [20,21].

Potholes, for example, are random excavations caused by wear and tear on the affected section of the road. If not attended to in time, they can cause further damage by collecting water, which accelerates wear and tear [22]. According to Staniek [23], road surface cracks in sections of roads supported by pillars can lead to regions falling off, posing a significant risk to human life and vehicles. The debonding process caused by the loss of strength in the adhesive used in road construction leads to structural degradation on roads [24]. The structural degradation identified as stripping is caused by the loss of bonds between solid aggregations of road construction material [25]. Stripping begins from the bottom layers of the roads and progresses upward, causing significant damage to the road surfaces. Ravelling of road surface happens when stripping starts on the upper layers and goes downward [26]. Road surface bleeding is another form of structural degradation on roads, which occurs when asphalt rises from the lower concrete layers to the surface layer of the road, leading to a shiny surface. The leading cause of bleeding on-road is hot weather, poor-quality asphalt, and low space air void content. Timely structural damage detection on roads supports taking necessary measures to repair or rebuild the damaged structures [27]. Regular assessments help to uphold motorists’ safety and save taxpayer money [14,28].

Scholars classify anomaly types into contextual, point, and collective anomalies [18]. Contextual anomalies are out-of-place objects such as fallen-off road cones [29] or animals on the road [30]. Point anomalies on the road refer to specific locations where unusual events or incidents occur, such as potholes or traffic accidents. Collective anomalies, on the other hand, refer to broader patterns or trends in road data that deviate significantly from the norm, such as a sudden increase in traffic volume or a rise in the number of vehicle breakdowns. Common anomalies include unsecured objects and debris that fly out of vehicles involved in accidents [31], small obstacles often overlooked such as speed bumps [32] or abnormalities in road terrain overlay, affecting self-driving cars [33,34]. Figure 1 illustrates a collage of on-road hazards from around the globe that ARDAD systems can help to mitigate.

CV-based ARDAD systems mostly employ data-driven ML algorithms that are trained on captured data samples representing normal behaviour and the abnormal behaviour and characteristics of the surveillance scene. The process typically uses supervised, semi-supervised, or unsupervised learning [35]. In other words, the ARDAD methods use visual observation that depends on the surveillance scene’s behaviour and characteristics. Hence, ML algorithms’ performance also depends on data supplied for training.

As the survey’s third contribution, we summarise the most popular publicly available datasets.

Due to the dynamic nature of road surveillance, ARDAD systems require expert feedback in the form of expert labelling or categorising of data into finite sets, such as roadside anomalies and defects (Figure 1), which could also result in re-training the model with an updated dataset. Supervised, semi-supervised or unsupervised learning are typically used in training such ARDAD frameworks [35]. Figure 2 illustrates the standard methodology for ARDAD system training and operations.

**Figure 2 sensors-23-05656-f002:**
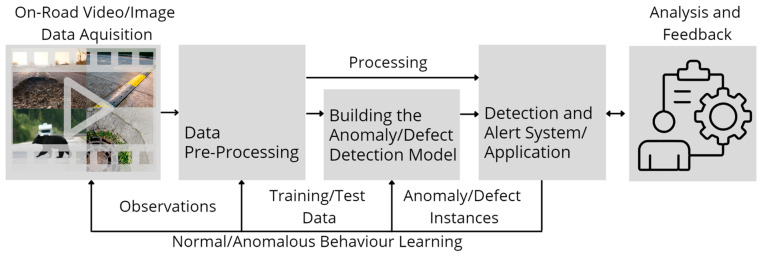
Overview of automated anomaly/defect detection process (derived from [36]).

Over the last few decades, the emergence of DL has brought the End-to-End (E2E) learning approach to the forefront of anomaly and defect detection modelling. Traditional ML models often rely on domain knowledge or domain experts to design or improve data pre-processing and feature extraction algorithms. E2E learning, on the other hand, reduces this dependency on expert knowledge and simplifies the process of extracting features or analysing discriminative properties from input data. Instead, the focus is on the input, such as an image vector, and the intended classification result from the system output [37]. In E2E learning, the model learns to extract invariant road features, recognise anomalies, or extract different surface textures in defect recognition.

As the fourth contribution, the systematic review reports on the popular machine and DL approaches and their performance applied to ARDAD systems.

### 1.2. Motivation and Contribution

The motivation for this systematic review lies in the understanding that road defects and anomalies significantly impact traffic safety and the overall economy. In this systematic review, studies from 2000 to 2023 are selected to capture the evolution of ARDAD methods and technologies over the past two decades. The selected time frame covers crucial developments, including a mathematical morphological method at the turn of the millennium [38], automated anomaly detection a decade later [39], and sophisticated surveillance techniques employing UAV swarms by 2023 [40].

Identifying road defects and anomalies helps reduce drivers’ risks while supporting road maintenance [12]. ARDAD systems can play a significant role in augmenting visual surveillance to safeguard the public and private transportation of modern cities roads [41], sub-urban and rural roads [42,43], animal hazard-prone hinterlands such as wilderness roads [30], and avalanche-prone mountainous roads [44]. This systematic review is the first in which the authors summarise the performance and accuracy of hazard detection systems used in road infrastructure surveillance achieved globally. The review proposes perspectives on existing technology, explores anomaly detection methods of the past three decades, and presents examples of anomalies and methods applied to detect and predict various static/dynamic anomalies and defects (Figure 3).

**Figure 3 sensors-23-05656-f003:**
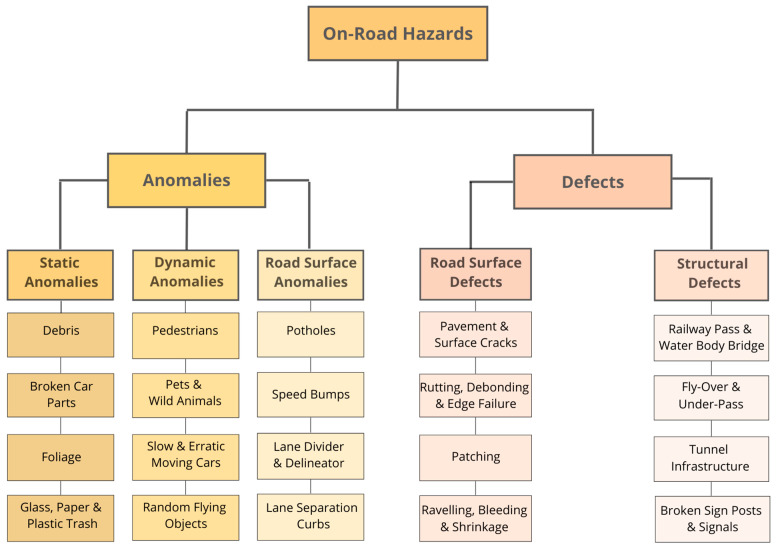
Types and sub-types of on-road hazard categories in the context of anomaly and defect detection using computer vision.

The survey analysed various processes based on environmental representation, features, approaches, and ML models. The systematic review’s contributions are listed as follows:Selection criteria and resulting review of globally relevant articles uniquely combining automated road defects and anomalies (ARDAD) peer-reviewed research since 2000.Discovery of the upward and exponentially growing trend of ARDAD surveillance automation since 2000.Taxonomy of machine and DL approaches combined with CV, including data acquisition technology and algorithms.List of popular and current open access ARDAD datasets.Critical analysis of the current state-of-the-art ARDAD methods to highlight the shortcomings that could be addressed in future research, including increasing environmental awareness of connected/self-driving cars.Compliance list adopted from the Preferred Reporting Items for Systematic reviews and Meta-Analyses (PRISMA) (http://prisma-statement.org, accessed on 20 December 2022) and applied to the ARDAD research context.

The introduction section of this systematic review discusses the significance of ARDAD methods and their development over the past two decades. Emphasis is placed on the role of sensor technology, computer vision, and ML techniques in enhancing traffic safety. The growing trend in surveillance automation is highlighted as a premise for the upcoming sections focusing on the systematic review approach, dataset analysis, and a critical analysis of ARDAD methods.

## 2. Research Questions and Review Approach

According to the systematic literature review guidelines [45,46,47], screening on-road anomalies and defects addresses a problem that can be prevented by detection, leading to the genesis of screening or intervention-type research questions. Furthermore, since the problem’s solution also depends on early problem detection, the research questions address the “preventive screening” problem. The research questions’ scope should be balanced so as not to be too specific or too broad. A well-formulated question determines (a) the criteria used to select studies, (b) the development of the search strategy; (c) the data to be extracted. The research questions answered by the systematic review are as follows:What are the best ML methods for improving classification performance and creating a robust detection and alert system?What implications does the up-to-date research have on motorists’ safety and future applications to related contexts, such as improving the environmental awareness of connected/self-driving cars?

The review process draws on empirical evidence from previous experiments, data collection, and studies. Figure 4 illustrates (a) literature review types and (b) how a systematic review of similar studies uses specific methods to identify, select, appraise, and synthesise the results.

### Data Gathering and Inclusion–Exclusion Criteria

The leading search engines used during data-gathering are Scopus and Google Scholar, linked by journal article search and subscription-based access from Auckland University of Technology’s (AUT) library (Table 1). The Boolean search (1) for article selection includes default settings for analysing titles, keywords, and abstracts.

The criteria (Table 1) are defined and aligned with the research focus. The articles were selected according to Equation (1). Once the refinement process was completed, a total of 195 articles were excluded, and from the selected papers, 48 deal with structural damage detection, another 47 deal with anomaly detection and the remaining 21 surveys. In Figure 5, the scatter-plot distribution shows the relationship of the number of articles reviewed on the various types of anomalies from January 2000 to May 2023 (i.e., date of publication).
((anomaly ∨ defect ∨ crack) ∧ (road ∨ motorway) ∧(prediction ∨ classification ∨ detection) ∧ (year ≥ 2000))(1)

A comprehensive selection process was conducted to identify relevant papers for further review within the scope of this study. Out of the initial pool of 311 papers, a total of 116 papers were chosen, which included 21 surveys and reviews (Figure 5). The remaining six papers consisted of reports, citations to research tools, or other types of valuable evidence that supported the review process. The criteria for advancing papers to the subsequent review stage were established based on the predefined guidelines outlined in Table 1.

Analysing the scatterplot data, we can observe approximate 18-month gaps between peaks and a notable increase in publications during 2020, followed by a decline in subsequent years. However, it is vital to provide a more insightful interpretation considering the impact of pandemic-induced lockdowns during 2020–2022, which resulted in reduced traffic, data collection, and occurrences of road damage. Further research is needed to gain deeper insights into the underlying factors driving these trends in ARDAD.

## 3. Datasets Reviewed

The research community can benefit from appropriate datasets when evaluating their road anomaly and defect detection models. Table 2 briefly describes prevalent open-access datasets for on-road anomalies and defect detection, which have been experimented on in the related literature. The datasets cover on-road hazards from simple potholes to more complex tunnels, concrete bridge defects, to avalanche debris flow affecting motorists’ safety. The motivation for assembling this diverse dataset is to provide unrestricted access without login requirements or paywall barriers. This section reports a selection of single-point, one-click access routes to frequently downloaded open access datasets for the research community (Table 2). Our goal is to promote inclusivity and remove possible discrimination, ensuring that researchers from all backgrounds can contribute to and benefit from the advancements in the field of ARDAD. To verify unrestricted, all-inclusive access and to promote privacy, we have tested the dataset access to ensure that all data are freely accessible without needing a login or being restricted by paywalls.

In summary, Table 2 provides an exhaustive review of 18 selected open access datasets pertinent to anomaly and defect detection, including potholes, debris flow, animals on the road, tunnel defects, and concrete bridge defects. As added parameters, each dataset is characterised by its purpose, configuration, origin or citation, strengths, limitations, and usage statistics.

Apart from the datasets provided in Table 2, the systematic review inspected datasets used by the studies that are not open access. This leads to identifying datasets available upon request or needing a paid subscription. For instance, the research on pavement crack detection [48] makes the CFD dataset, Crack500 dataset, and a customised dataset called CrackSC available on request. In another study [49], a wide variety of road obstacle datasets are available on request. The road anomaly detection study [50] provides multiple datasets; however, login access is needed to download them. The research on the Adaboost algorithm for pavement distress detection [51] provides access to the dataset through the journal’s website for readers with paper access. The research on thermal image analysis for defect detection [52] provided the dataset upon request.

## 4. Literature Review

The review summarises perspectives on existing detection technologies and presents examples of methods developed since 2000 for applications to detect and predict static/dynamic anomalies and defects. Unlike the subjective nature of topic-oriented narrative literature reviews, the systematic literature review approach represents an opportunity for repeatable article selection and synthesis of follow-up reviews. As no detection system can be applied globally, Figure 3 illustrates a review-based breakdown of on-road hazards to motorist safety.

Image processing-based ARDAD systems play a significant role in enhancing traffic safety through visual surveillance [53]. Images of road sections are taken and analysed to detect structural variations and anomalies from time to time. In addition to image processing, CV combines artificial intelligence (AI) approaches to derive meaningful information from images and videos [54]. When merged with Global Positioning Systems (GPS), telescopes, binoculars, closed-circuit television (CCTV), vehicle-mounted video recorders and cameras, and low-cost mobile cameras, image processing-based visual surveillance can significantly increase the efficiency of ARDAD systems [18,40,55,56,57]. Maya et al. [58] proposed a delayed long short-term memory (dLSTM)-based technique that is trained in a normal state and predicts abnormalities depending on the m-score defined in Equation (2). Here, the m-score is the normalised anomaly score *R*(*t*) within the abnormal state, and if in a dataset, the *T*_2_ anomaly occurs at time *t*_1_, the resultant m-score value is as follows:(2)$$m\text{-}\mathrm{score}={median}_{i\in \left[t_{1},t_{1}+1,\ldots ,T_{2}\right]}R\left(i\right)$$

Based on expert feedback, the anomaly is detected if the m-score is above the set threshold. The method is reported to be flexible when combined with other anomaly prediction models. The U.S. Department of Transportation (USDOT) devised a convention to rank road surface distress [59], as shown in Equation (3). The pavement condition index (PCI) is generated using a weighted sum of surface condition rating (SCR) and roughness condition index (RCI).
(3)PCI=0.6·SCR+0.4·RCI

Structural damage on roads is caused by thermal action, external conditions and physical strain exerted by vehicles. Various anomalies and surface defects such as cracks and potholes are caused mainly by deformities such as debonding, stripping, ravelling, bleeding, shrinkage of road layers and swelling of road layers [60,61]. A wide range of datasets are produced to derive meaning from images of surfaces with such defects and identify the underlying conditions of the roads. According to Bhatt et al. [10], three distress categories are used to classify anomalies: cracking, visco-plastic deformations, and surface defects.

Thus, the current research mainly focuses on surface damage detection, anomaly detection, analysis and prediction using computer vision in association with traditional ML and DL technologies. The literature review of surveys between 2000 and 2023 shows that most of these can be classified into road surface defects or on-road anomalies (Table 3).

Image processing based on traditional methods (statistical and classical ML) has been used to analyse road sections’ images to detect defects [10]. Examples of applied methods in image processing include logical and linear regression [60], naïve Bayes [71], support vector machine (SVM) [57], random forest (RF) [61] and more. Statistical and traditional ML-based image processing might be inefficient due to known difficulties in handling noise in previously unseen images, complex textures in different backgrounds or variations in lighting conditions of surfaces. The shortcomings of statistical and traditional ML motivated researchers to investigate new approaches. Other anomaly and defect detection development suggests the following three main approaches: feature extraction-based image processing using DL, ML and ensemble learning models [72,73]. Different works have also used 3D imaging and LiDAR-based anomaly and defect detection methods [19].

### 4.1. ML-Based ARDAD

Li et al. [74] developed the defects detection and localisation network (DDLNet), a vision-based method for detecting, classifying, and geolocating defects using region-growing, edge detection, and threshold segmentation techniques. The DDLNet achieved 80.7% detection and 86% localisation accuracy. Cha et al. [75] proposed the utilisation of traditional Canny and Sobel edge detection methods, achieving an impressive accuracy of 98% in detecting block edges, edge cracks, and longitudinal and transverse cracks. For visco-plastic deformations, edge detection can efficiently detect pothole edges [76], depressions, stripping, and ravelling with high accuracy of 99.11% [77]. In some cases, edge detection is achieved using Prewitt, Canny, and Sobel operators [78]. Each operator impacts the detected edges within an image differently based on each operator’s ability. Chatterjee and Saeedfar [79] proposed an improved Canny edge detection method that incorporates genetic algorithms and enhances the blurred edges using the Mallat Wavelet transform with an average detection accuracy of 91%. Vigneshwar et al. [80] proposed the binary conversion of greyscale images for anomaly detection. They set a threshold, compared pixels for background or target area identification, and used threshold segmentation, edge detection, and K-Means clustering for crack and defect detection with an average accuracy of 80.60%, 90.19%, and 82.47%, respectively.

Table 4 presents a comprehensive overview of recent studies employing ML algorithms to detect road anomalies, showcasing a range of accuracy rates between 86.3% and 97.8%. While these studies significantly contribute to the advancement of traffic safety, autonomous driving, and urban planning, critical evaluation reveals limitations. These limitations include concerns about data accuracy, constrained feature applicability, and challenges in domain adaptation, which warrant further investigation and development.

The AdaBoost algorithm, proposed by Wang et al. [51], utilises supervised data for detecting surface defects such as ravelling and bleeding. The algorithm consists of a decision tree with elements categorised as root, leaf, and decision nodes. The collected data are passed through the root node and classified at each layer of the decision tree until it cannot be further classified. The sample data are divided into subsets for precise and optimal classification results. Each subset of the training data is assigned a leaf node, which should also have an associated class. Fan et al. [84] proposed three different decision trees in the AdaBoost algorithm for detecting road surface defects. Among these, the C4.5 decision tree continually prunes leaf nodes and adopts the root node as the new leaf node. The CART decision tree’s pruning process, unlike that of C4.5, uses a verification data set to prevent data overfitting. The ID3 decision tree calculates the maximum gain of all the sample value data and assigns features to nodes. The recursive generation of decision trees occurs using these features as leaf nodes [85].

Feature extraction-based ML methods are considered advantageous for their simplicity, according to Avci et al. [13]. By performing feature extraction and classification, these techniques can be made more generic and effective in detecting structural damage. Hoang and Nguyen have developed various ML methods to detect different classes of static anomalies and structural damages within roads [86] that include support vector machines (SVM), random forest (RF), and artificial neural networks (ANN) using both labelled and unlabeled datasets. The supervised approaches relying on labelled training data in road anomaly detection are naïve Bayesian, RF, SVM, ANN, and logistic regression. Artificial neural networks and RF facilitate efficient visco-plastic deformations and cracking defect detection. Fakhri and Saadatseresht [87] proposed a model based on the random-forest supervised data model to detect cracks whilst overcoming the challenge of uneven edges of the cracks and cracks existing in complex topologies. Table 5 highlights the evolution of ARDAD methods, demonstrating a shift from traditional ML and statistical techniques to DL approaches. While both categories have contributed to automating defect detection and enhancing road safety, they exhibit limitations such as dependency on image quality and environmental conditions.

In detecting on-road anomalies, unsupervised learning models hold potential advantages as they do not rely on labelled data for sample classification, unlike supervised learning models, which depend on subjective human input [93,94]. As a result, the output of unsupervised learning models is not predetermined, allowing computers to independently discern anomalies in the data through classification processes [95]. Ishtiak et al. [43] proposed a system for identifying and categorising various road conditions, including visco-plastic deformities and defects. This approach uses a statistical analysis method and a scoring function considering several factors, such as road colour, material, and image quality. Despite the model’s high accuracy, ranging from 77% to 89% across diverse road conditions, it has shown limitations in distinguishing shadows from road anomalies and analysing roads with water on the surface. Chatterjee et al. [79] proposed a machine-learning approach for crack detection, relying on feature extraction from image superpixels. The approach involves extracting 40 features, including variance, skewness, six Grey Level Co-occurrence Matrix (GLCM) features, and 32 Variance-of-Gabor (VoG) features. The study compared four classifiers, with gradient boosting (GB) being the most accurate at 92.77%, followed by random forest (RF), artificial neural network (ANN), and linear support vector machine (L-SVM). Naddaf-Sh et al. [96] proposed a novel model for detecting visco-plastic deformations and cracks, leveraging a multivariate statistical hypothesis and a minimum intensity path window for anomaly extraction. Despite a competitive F1 score of 56%, increased inference time during real-time prediction and transferred augmentation policies might hinder the model’s performance. Mahadevan et al. [97] proposed a model that detects abnormalities in crowded scenes by considering temporal and spatial normalcy using a mixture of dynamic textures. The algorithms tested in the study show varying performance (25% to 42%) regarding an equal error rate and anomaly localisation, with MDT outperforming the others with a detection rate of 45%. Table 6 presents diverse, evolving methodologies in road defect detection, from simple image processing to sophisticated deep-learning models. These research efforts have led to accuracy rates ranging from 54% to over 99%, indicating a promising trend in the field. These studies collectively demonstrate evolving methodologies, from simple image processing to sophisticated deep-learning models.

Table 6 provides a comprehensive overview of road defect detection and classification research, offering a roadmap for further advancements towards safer and more efficient transportation systems.

A computer vision-based approach by Cha et al. [75] summarised that DL, as a powerful approach for object detection, image segmentation, and classification, has been used to detect anomalies and defects such as cracks, surface defects, visco-plastic deformations, and traffic anomalies. As a case in point, their CNN-based approach achieved accuracies of 98.22% out of 32K images and 97.95% out of 8K images in training and validation, respectively. The proposed CNN method showed very robust performance compared to traditional edge detection methods. Opara et al. [71] proposed a DL approach involving binary and multi-class classifications to detect anomalies in the RGB images (2400 × 2000 pixels) with a high F1 value of approximately 60% at 18,000 iterations. The study utilised a loss function that included terms for localisation, confidence, and classification errors to detect objects more accurately and effectively. Non-maximum suppression was applied to select the appropriate bounding box from the many predictions. On the other hand, multi-class classification is suggested for analysing road sections with multiple anomalies. At the same time, the authors also provided insights on performance trade-offs by adjusting hyperparameters and achieved state-of-the-art performance with an F1 score of up to 94.4% on three benchmark datasets [102].

The pixel segmentation method for pavement damage detection using a thermal-RGB fusion image-based model achieved high accuracy with a pre-trained EfficientNet B4 backbone architecture and an argument dataset with a detection accuracy of up to 98.34% [52]. To detect visco-plastic deformation, surface defects and cracks, Minhas et al. [103] proposed an efficient pixel segmentation model (F1 score 0.89) with four convolutional layers, three layers for segmenting the sample image directly connected to the input, and two for maximum pooling. To achieve optimal results, pixel segmentation models are divided into decoder and encoder layers [104]. The encoder layer is used to map the image features, while the decoder layer establishes feature vectors of images during the segmentation process. The decoder layer also develops a probability distribution of every pixel identified within the images. However, the object detection approach, which identifies and binds objects with boxes within captured images, has limited usability. Table 7 summarises various studies, including wildlife–vehicle collision analysis, pothole detection, road surface monitoring, and anomaly detection for autonomous vehicles. While the studies propose different methods ranging from traditional statistical analysis to ML and edge AI-based approaches, each method has limitations, including limited data, reliance on manual labelling, lack of road roughness estimation, and potential false positives. Nonetheless, these studies demonstrate the potential for technology to improve road safety and maintenance.

The selection of the appropriate neural network for a given problem depends on various factors, including the complexity of the intended solution, computing resources, and data availability [70]. Traditional ML methods can be advantageous when the dataset is small or limited, but their performance may plateau with more data. In contrast, deep neural networks tend to perform better with a large amount of data, enabling the identification of subtle dependencies through more dense layers. Oliveira and Correia [98] have reported that less sophisticated traditional machine-learning methods can be effective in the case of small datasets, particularly in dynamic anomaly detection systems. However, the performance of deep neural networks can improve with more data and complex architectures [37].

Neural networks’ complexity increases with the need to process large amounts of data. Shallow neural networks typically have fewer layers and may not use backpropagation algorithms. However, deep neural networks usually perform better with enough data and sufficient computing resources than the traditional approach. However, according to Cui et al. [107], traditional machine-learning methods, such as support vector machines, usually perform better at anomaly detection and generally require fewer computing resources for data processing.

Due to their high efficiency in local filtering, noise detection, and overall transforming domain and non-local mean filters, convolutional neural networks (CNNs) have been increasingly used in anomaly detection and denoising images from sections containing anomalies [108]. Akagic et al. [109] proposed a two-step CNN model for road anomaly detection. Different images are fed into 32 by 32 CNN layers during the first step to train the model. Greyscaling is performed, followed by thresholding to detect identifiable anomalies within the image. According to Chambon and Moliard [28], CNN datasets are trained using various data such as different target anomaly types, road width, weather and lighting condition patterns, condition of the road surface, and the height of elevated road supports such as pillars of natural elevators. In another study by Li et al. [74], a Deep Dual Localisation Network (DDLNet) is proposed for defects detection and geolocalisation in a unified model. The model combines a novel defects RPN and a NetVLAD module for detection and geolocalisation. The authors also propose a novel data augmentation method and hard negative mining strategy to improve detection accuracy and reduce the possibility of triggering false alarms.

The twice-threshold segmentation method demonstrates higher accuracy of up to 98% in detecting cracks in runway images containing road markings, outperforming traditional threshold segmentation algorithms such as Otsu (40%) while maintaining adaptability for various applications [110]. Amhaz and Chambon [100] proposed the Minimum Path Selection (MPS) algorithm for crack detection with a Dice Similarity Coefficient (DSC) of 0.77 on 2D pavement images. However, further advancements in computation time and adaptability to 3D imaging systems are necessary for broader applications. The Dijkstra algorithm is then used to estimate the minimal path between the two points, which can be manually corrected if a false minimal path intersects with the crack. The post-processing method is then applied to estimate the crack’s thickness and provide the complete crack pattern. Shankar and Wang [111] proposed a Fully Convolutional Neural Network (FCNN) model for anomaly detection, while Ishtiak and Ahmed [43] utilised a two-step image classification approach in their FCNN model. The first step involved feeding road surface images into the FCNN, with the model achieving 87% accuracy for all classes. In the second step, the model was trained with threshold images to establish cutoff levels for anomalies and structural damage detection.

### 4.2. Ensemble Learning for Improved Anomaly and Defect Detection

Doshi and Yilmaz [112] propose ensemble learning to improve the efficiency of different ML approaches used in static anomalies and structural damage detection. The ensemble model (EM) approach uses various trained models to predict the three proposed static anomaly approaches. The EM uses a variety of trained models for static anomaly prediction. Ensemble learning improves the accuracy of training the ML models. The ensemble prediction (EP) approach utilises images generated from the test time augmentation (TTA) and ensembles the anomaly predictions derived from these images. The hybrid approach uses EM and EP models to conduct anomaly predictions. Hegde et al. [42] proposed a DL approach for road damage detection and classification using YOLO and ensemble learning, achieving an F1 score of up to 0.67, demonstrating the potential of these methods for smart city applications.

Alipour et al. [83] investigated the use of ensemble learning for crack detection and proposed a method that combines pre-trained models developed for specific types of materials. To achieve this, the softmax operator was utilised to extract the probability of each prediction, where $S_{j}\left(x\right)$ represents the observation probability of class *j*, and *n_class* is equal to two for the binary crack vs. non-crack problem. The proposed method leverages the knowledge stored in both material-specific models to make a single prediction for each future image regardless of the material. The softmax operator is shown in Equation (4), where the denominator’s last variable l in the exponent $e^{xl}$ represents the class label.
(4)$$S_{j}\left(x\right)=\frac{e_{j}^{x}}{\sum_{l=1}^{n\_class}e^{xl}}$$

A hybrid algorithm, such as the non-maximum suppression (NMS) algorithm, derives a single output from these outputs [42]. The algorithm works by filtering out the overlapping or duplicate predictions from the predictions pool. All the images captured from road surfaces are then passed through the models for state prediction by applying the NMS. In EMs, the one-stage detector models include the ultralytics-You Only Look Once (u-YOLO) model, so it is possible to combine various u-YOLO models. In order to train a u-YOLO model, different input parameters to these models are tuned [113]. Different trained models are achieved by selecting different combinations of data for tuning. A favourable subset of these models is chosen for use, although the choice is based on the available training data in such cases. All the images captured from road surfaces are then passed through the models for state prediction by applying the NMS. Ensemble learning significantly reduces the prediction variance, making the approach highly accurate. The hybrid approach applies the EP model approach to each EM model. After every test image has been transformed through TTA, each EM model is given an input of the augmented images. The models output bounding boxes once NMS is applied to derive a prediction. The corresponding structural damage or anomaly on the road section is determined from the bounding boxes.

Based on the availability of computing resources and data volume, both ML and DL have their uses and potential for future technologies (Table 8).

### 4.3. Detection Based on 3D Imaging Methods

Traditional anomaly detection methods have predominantly relied on 2D imaging techniques, such as Bidimensional Empirical Mode Decomposition (BEMD), used for pavement crack detection [114]. However, with the development of range-based sensors and stereo cameras, 3D imaging methods have become more efficient. In addition, 3D stereo vision is particularly effective in estimating the depths of cracks and visco-plastic deformations with a precision score of up to 90% [115]. Microsoft Kinect and laser-imaging techniques are used in traditional methods and DL neural networks as a new research direction, including CrackNet, CrackNet II, and CrackNet V.

Table 9 shows that these methods have demonstrated significant potential in object recognition, pose estimation, and autonomous navigation applications. However, the accuracy and reliability of 3D imaging methods heavily depend on factors such as sensor resolution, calibration accuracy, and environmental conditions. Nonetheless, the continual development of 3D imaging technologies presents promising opportunities for enhancing the capabilities of various applications in fields such as robotics, autonomous driving, and industrial automation.

Chen et al. [30] proposed a cost-effective approach for detecting deer crossing roads using 360° LiDAR sensors. The proposed algorithm can detect deer with a maximum radius of 37.74 m around the LiDAR sensor, which can trigger warning signs for drivers. While the method has limitations in detecting small animals and tracking individual deer, it shows promise for improving traffic safety and analysing wildlife behaviour.

Zhang et al. [116] proposed a CNN model that utilises 3D imaging methods that represent different 3D view data of images into one compact shape descriptor. Such models extract 3D data from the images and pass it to an ML model to detect anomalies and structural damages [102]. The 3D data are then used to train classifiers. The spatial information of road surfaces, such as width, length and depth, is represented by the 3D data [121]. Medina et al. [122] proposed a 3D imaging method based on laser imaging that models road surfaces using dense networks of 3D points.

Frequency analysis, mostly Fourier transformation, is applied to distinguish between the different anomalies. Akarsu et al. [101] proposed an improved Fourier transformation model such that the model takes into account non-uniform illuminations on surfaces. The method differs from the mentioned method because it utilises probabilistic relaxation and is the only effective 3D imaging method to detect road surface defects such as bleeding and ravelling. Furthermore, it also detects likely occurrences of visco-plastic deformations and cracks.

Figure 6 depicts the volume of literature reviewed based on detection origins divided by the ML methods’ taxonomy. Deep learning (34%) is the most popular method, followed by traditional ML (26%) and ensemble learning (26%). In comparison, 3D image-based techniques (14%) are the least represented by the reviewed ARDAD systems. Considering the timeframe of the literature reviewed, recent advancements in ML techniques may impact the overall taxonomy distribution and the road defect detection landscape. The growing popularity of DL approaches is likely due to their ability to process large datasets and automatically extract relevant features. However, traditional ML and ensemble learning methods are still widely used across ARDAD systems. However, in discussing the benefits and drawbacks of each taxonomy, it is essential to acknowledge the gaps in the literature and encourage further research to explore underrepresented ML methods or road defect types.

## 5. Gaps, Challenges, and Limitations

The road surveillance research domain is highly dynamic; the road surface and supporting infrastructure defects do not appear in uniform shapes or sizes, nor do the anomalies follow a uniform pattern, which leads to multiple challenges. An example of a significant gap and future opportunity is that the current detection methods do not evaluate or provide implications on how the defect or anomaly can directly affect motorists’ safety.

While the review provides insights into the state-of-the-art ARDAD methods, it has some limitations. First, the review primarily covers peer-reviewed articles written in English, which may exclude valuable information from other sources such as technical reports, some conference proceedings, commercial product documentations and patents. Second, the inclusion and exclusion criteria rely on ARDAD-associated terminology and concepts, which may not include relevant studies that use different terminology or naming conventions. To address these limitations, future reviews could consider broadening the search criteria to include additional sources of information and exploring alternative terminologies or approaches.

Reportedly, supervised techniques usually perform better when labelled data are available [69] because using labelled data during training allows supervised learning methods to detect boundaries and classify normal or anomalous classes. However, sometimes the training data do not include all types of anomalies, which leads to supervised approaches overfitting and performing poorly on new anomaly data. Hence, the availability of labelled anomaly data (or rather lack of it) creates an opportunity for applications and advancements in semi-supervised and unsupervised ML techniques. To address this observation, within the scope of this survey, we have provided a list of frequently downloaded open access on-road anomalies and defect image datasets.

Developing a robust ARDAD system is challenging; when summing up the body of literature on the topic, the main challenges and opportunities are as follows:Despite setting the inclusion parameters for publishing dates between 2000 and 2023, the literature search yielded only 311 papers. Due to the focused selection criteria, the systematic review included only 116 papers (Table 1).Contrary to our expectations, the number of computer vision-based studies directly impacting motorist safety was lower than expected.

For research replication, we adapted the PRISMA (http://prisma-statement.org, accessed on 20 December 2022) checklist, which is common for systematic reviews in health science. The adapted PRISMA checklist extension is important for future systematic reviews of ARDAD (and similar CV contexts). The PRISMA checklist extension is provided in the Appendix A.

## 6. Conclusions and Future Work

Motivated by the need to accelerate technological advancements that can improve traffic safety and reduce incidents, this systematic review analyses the literature on automated road defects and anomaly detection (ARDAD) systems from 2000 to 2023. As a result, the systematic review covers peer-reviewed articles (N = 116) associated with types of roadside anomalies and defects that are jointly intended to help prevent the loss of lives, injuries and infrastructure damage, ensuring on-road and structural integrity.

In the context of augmenting on-road surveillance for ease of maintenance, such as structural damage detection and hazard prevention via predictive monitoring, the review summarises the ARDAD methods, including the achieved performance using traditional ML, and DL, combined with sensor technology. Notably, it quantifies the achieved performance of these methods, providing insights into their effectiveness. Additionally, the review provides a taxonomy of ARDAD methods and descriptions, including a list of frequently downloaded open access on-road anomalies and defect image datasets (D = 18), facilitating future research and benchmarking.

Considering the current publication trends, the advancements in video technology, availability of sensors and computing resources in general, there is an exponential growth in ARDAD research publications from 2000 to the present day. As anomaly detection intersects with automatic road traffic surveillance, this survey can also be a valuable resource for interested researchers working on related contexts.

Due to the impact of the global pandemic and lockdowns from 2020 to 2022, there was less traffic and opportunities for new data collection compared to the previous years. The exponentially growing trend in the number of research publications during the period from 2015 to 2020 could be explained by earlier data collections prior to the global pandemic (Figure 5). In the authors’ view, the growing trend surrounding ARDAD technologies and research is likely to reach its peak, aligning itself with the “Innovation Trigger” stage of “Gartner’s technology adoption hype cycle framework” [123]. As such, future work on ARDAD technologies is likely to consider Gartner’s framework for a better understanding of a current project position on the hype cycle, to project the adaptation and maturity levels (of ARDAD technologies), to identify practical aspects of technology transfer such as self-driving vehicles and to identify the possible impact on society.

Considering the state-of-the-art ARDAD methods, we conclude that the latest IoT, 5G and 6G communication technologies, swarm drones, satellite imagery, cloud computing and GPS have the potential for near-future research and further expansion of related research contexts. The benefits of ARDAD methods to humanity include the utilisation and advancements of AI, CV, and semi and self-learning techniques to support intelligent vehicles, urban planning, intelligent transportation systems, connected or self-driving vehicles, improved road surveillance, reduced road maintenance costs, and increased traffic safety.

In order to enhance future research in the field of ARDAD systems, there is a crucial need for more comprehensive performance/meta-analyses that can evaluate the efficacy and efficiency of various ARDAD methods in real-world settings. While not a full meta-analysis, our systematic review provides a strong foundation, serving as a platform for future research. This potential conversion would enable quantitative data synthesis, further advancing our understanding of ARDAD technologies and facilitating evidence-based decision-making. Additionally, quantifying the societal and stakeholder impacts resulting from the implementation of ARDAD systems would offer valuable insights for policymakers and industry professionals.

Overall, this systematic review is a significant milestone in ARDAD systems, bridging a crucial research gap with its comprehensive analysis of traffic hazards ranging from urban cities to the wild hinterlands. Our commitment to inclusivity is evident in examining often-overlooked road hazards such as avalanches or cattle on the road, showcasing our genuine belief in uncovering hidden knowledge from future data or previously unseen or untested datasets. This systematic review establishes a foundation for future research endeavours in ARDAD systems and highlights the potential of emerging technologies to drive advancements in traffic safety and road maintenance. Our research findings inspire optimism based on emerging technologies’ potential to facilitate advancements aimed at improving safety and saving lives and making a positive impact on global society.

## Figures and Tables

**Figure 1 sensors-23-05656-f001:**
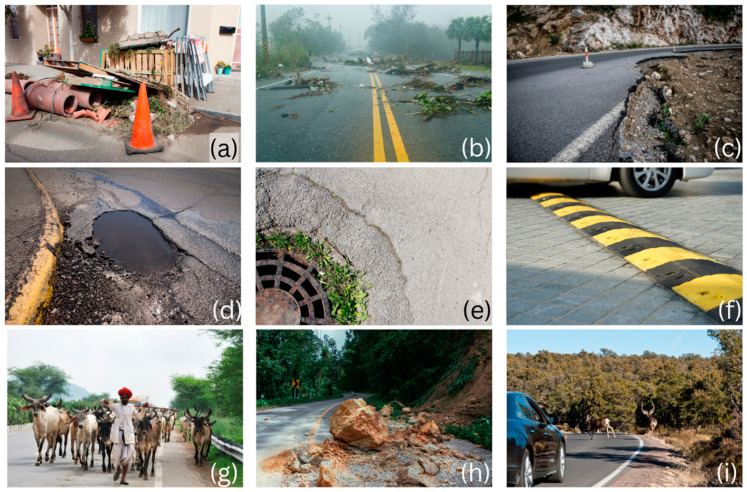
Various types of roadside anomalies and defects, including (**a**) road maintenance objects and construction debris, (**b**) debris fallen on-road, (**c**) road surface failure, (**d**) potholes, (**e**) maintenance holes and pseudo potholes, (**f**) speed bumps, (**g**) farm animals on the road, a common on-road hazard type, (**h**) landslide debris, and (**i**) wild animals jumping in front of a speeding car. (Source: All pictures reproduced under paid licence with Canva, www.canva.com, accessed on 10 January 2023).

**Figure 4 sensors-23-05656-f004:**
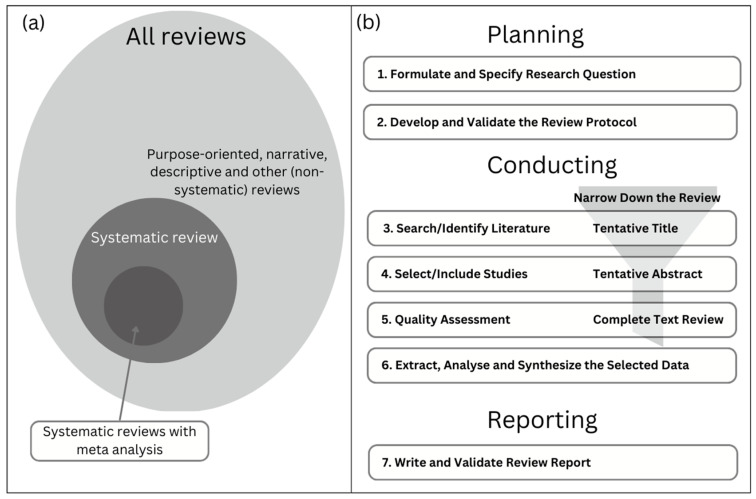
Article selection and processing workflow: (**a**) taxonomy of literature reviews, (**b**) review process used in the study (adapted from [46,47]).

**Figure 5 sensors-23-05656-f005:**
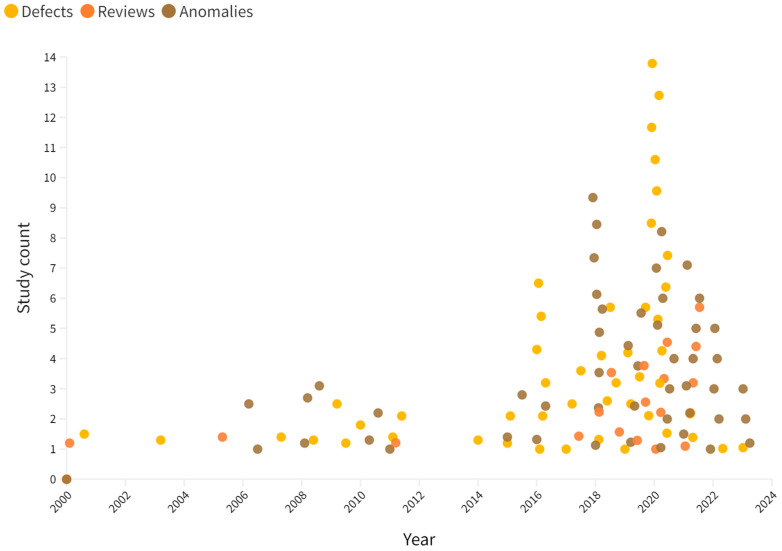
The number of articles reviewed on various types of on-road anomalies and defects since 2000, highlighting the growing research interest in enabling technological advancements. Note that the data presented are based on our selection criteria, which focus on specific aspects of ARDAD research.

**Figure 6 sensors-23-05656-f006:**
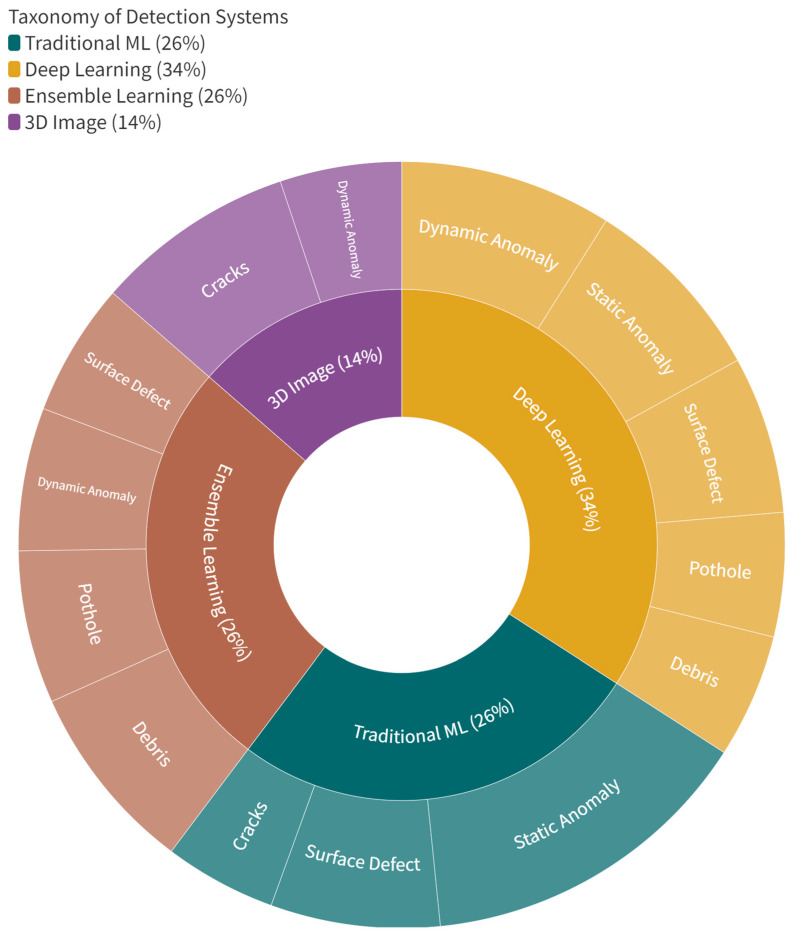
Volume of literature reviewed based on origins of detection divided by the taxonomy of ML methods.

**Table 1 sensors-23-05656-t001:** Inclusion–exclusion criteria settings.

Criteria	Included	Excluded
Date	Include the studies between 2000 and 2023 depending on the topic and availability	Exclude older versions if new versions of studies are available
Topic	Studies that focus on ML in the context of ARDAD systems for motorist safety	Exclude studies that do not address detection in terms of computer vision or motorist safety
Source	Scopus, IEEE, Science Direct, DOAJ, and Google Scholar	Web search of non-peer-reviewed sources, non-English publications and non-scholarly work sources.
Peer review	Peer-reviewed conference papers, journal articles, technical reports, and web-based articles important to research questions	Studies include dissertations, thesis, posters, short papers, and abstracts.
Research/study design	Studies focusing on video and image processing, visual defect/anomaly detection for motorist safety	Studies that do not deal with video and image processing
Setting	Outdoor/indoor conditions with varying lighting	Permanent backgrounds or unvarying lighting conditions in surveillance scenes
Reported Outcomes	Precise classification and detection outcome with a reasonable success rate	Unclear outcomes and below-average success rate of detection accuracy

**Table 2 sensors-23-05656-t002:** Frequently downloaded open access on-road anomalies and defect image datasets (URLs in the table accessed on 4 June 2023).

No.	Dataset Link	Purpose	Configuration
1	Road surface anomalies kaggle.com/datasets/aminumusa/road-dataset	Detect potholes and improve road maintenance; originated from Nigerian highways	789 good surfaces, 670 potholes images, Uniform 256 × 256 pixels
		**Paper of Origin/Use**: https://tinyurl.com/ydbwp39f **Strength**: Clear distinction between classes, real-world images **Limitations**: Limited variety of images specific to Nigerian highways **Statistics**: 14 downloads and 4 citations	
2	Pothole image dataset kaggle.com/datasets/sachinpatel21/pothole-image-dataset	Detect potholes on roads	600+ .jpg images of the potholes, web scraped
		**Paper of Origin/Use**: https://ieeexplore.ieee.org/abstract/document/9824637 **Strength**: Diverse dataset with pothole images from varied road surfaces **Limitations**: The collection method may have resulted in noisy or duplicate images **Statistics**: 3454 downloads and 2 citations	
3	Debris flow data zenodo.org/record/6679461 datahub.hku.hk/articles/dataset/Dataset_and_supplementary_movies_for_geophysical_mass_flows_against_a_flexible_ring_net_barrier/20349192	Debris and rock avalanches	228 videos in total debris caused by an avalanche
		**Paper of Origin/Use**: https://agupubs.onlinelibrary.wiley.com/doi/abs/10.1029/2022JF006870 **Strength**: Detailed and high-quality data covers a range of scenarios **Limitations**: Some of the samples from the laboratory setting may not reflect real-world conditions **Statistics**: 145 downloads and 1 citation	
4	Pothole and road images kaggle.com/datasets/virenbr11/pothole-and-plain-rode-images	Road defects and pothole detection	740 images of road potholes
		**Paper of Origin/Use**: https://www.hindawi.com/journals/cin/2021/6262194/ **Strength**: A proper train-test split for potential applications in training ML models for road maintenance, traffic management, and autonomous vehicle navigation systems **Limitations**: Images scraped from the web may result in inconsistencies. **Statistics**: 1528 downloads and 3 citations	
5	On-road anomalies and obstacles segmentmeifyoucan.com/datasets	Providing pixel-level annotations for the classification of anomalies and other hazardous obstacles	467 labelled and unlabeled images of on-road anomalies and obstacles
		**Paper of Origin/Use**: https://datasets-benchmarks-proceedings.neurips.cc/paper/2021/hash/d67d8ab4f4c10bf22aa353e27879133c-Abstract-round2.html **Strength**: The dataset builds upon the popular Cityscapes dataset, making it useful for training ML models for anomaly detection and obstacle identification in urban scenes **Limitations**: Low anomaly diversity in datasets limits generalisation and may cause overfitting **Statistics**: Unknown number of downloads and 34 citations	
6	Speed hump/bump dataset data.mendeley.com/datasets/xt5bjdhy5g/1	Detecting and classifying speed humps/bumps in real-world conditions to improve real-time applications such as self-driving cars	Total 3000 .jpg images of humps/bumps with varying conditions, such as different types of humps/bumps, illumination conditions, and geographical locations
		**Paper of Origin/Use**: https://www.sciencedirect.com/science/article/pii/S1877050918320295 **Strength**: The dataset contains diverse speed humps and bumps under varied conditions, improving model generalisation **Limitations**: Limited public availability, geographical scope, manual labelling, and potential biases affecting model performance **Statistics**: 4530 downloads and 46 citations	
7	Crack-forest dataset github.com/cuilimeng/CrackForest-dataset kaggle.com/datasets/mahendrachouhanml/crackforest	Annotated road crack image database for developing and evaluating automatic road crack detection algorithms	Collection of 118 annotated images with ground truth labelling of cracks and background pixels, used for training and testing crack detection models
		**Paper of Origin/Use**: https://ieeexplore.ieee.org/abstract/document/7471507 **Strength**: A diverse set of annotated road crack images randomly shuffled with 80%-20% splits, respectively, for training and testing crack detection models **Limitations**: Limited number of images **Statistics**: 423 downloads and 749 citations	
8	Pothole dataset drive.google.com/drive/folders/1vUmCvdW3-2lMrhsMbXdMWeLcEz__Ocuy https://www.kaggle.com/datasets/felipemuller5/nienaber-potholes-1-simplex	Pothole detection	Two sets of 650 annotated pothole images, with variations in complexity and some overlapping files
		**Paper of Origin/Use**: https://ieeexplore.ieee.org/abstract/document/7376642 **Strength**: Realistic pothole images with varying real-world scenarios and comprehensive annotations in two papers for pothole detection models **Limitations**: The possibility of duplicate image names can be problematic **Statistics**: 645 downloads and 102 citations	
9	Road damage dataset paperswithcode.com/dataset/rdd-2020 data.mendeley.com/datasets/5ty2wb6gvg/1	Damaged road surface detection	A total of 26,620 .jpg images of 31,000 instances of road damage from multiple countries using smartphones
		**Paper of Origin/Use**: https://arxiv.org/abs/2008.13101 **Strength**: A large, diverse dataset with annotations helpful in developing deep learning models **Limitations**: NA **Statistics**: 2346 downloads and 76 citations	
10	Road anomalies epfl.ch/labs/cvlab/data/road-anomaly	Dynamic anomaly detection	120 images with associated per-pixel labelled unusual on-road entities such as animals, rocks, traffic cones and other obstacles
		**Paper of Origin/Use**: https://arxiv.org/abs/2008.13101 **Strength**: Realistic representation of road hazards, per-pixel labels for training, and the benchmark for evaluation **Limitations**: The dataset is designed for a specific purpose and may not be suitable for other applications or research topics **Statistics**: Unknown number of downloads and 84 citations	
11	Road surface potholes sites.google.com/view/pothole-600/dataset	Pothole detection and classification	600 RGB images and pixel-level annotations collected using a ZED stereo camera; the road disparity images were estimated using Perspective Transformation—Search Range Propagation (PT-SRP)
		**Paper of Origin/Use**: https://link.springer.com/chapter/10.1007/978-3-030-66823-5_17 **Strength**: Contains annotated images that can be used for training and testing pothole detection algorithms. Stereo camera use allows for the estimation of disparity images, which helps improve the accuracy of pothole detection **Limitations**: The dataset was collected using a single camera setup, which may limit its generalisability to other camera setups **Statistics**: Unknown number of downloads and 28 citations	
12	Labelled pothole dataset public.roboflow.com/object-detection/pothole kaggle.com/datasets/chitholian/annotated-potholes-dataset	Fully annotated image dataset for pothole detection	665 images with a total of 1740 annotated potholes. 532 (80%) training images, 133 (20%) test images.
		**Paper of Origin/Use**: https://link.springer.com/chapter/10.1007/978-981-16-6636-0_44 **Strength**: It is a fully bounding box with annotated images of potholes and damaged roads **Limitations**: 57.1% of images are web scraped, potentially consisting of duplicate images **Statistics**: 2222 downloads and 1 citation	
13	Pothole detection dataset kaggle.com/datasets/atulyakumar98/pothole-detection-dataset	Road surface pothole detection	352 undamaged road images and 329 pothole images
		**Paper of Origin/Use**: https://ieeexplore.ieee.org/abstract/document/9850988 **Strength**: Diverse dataset with annotations helpful in developing deep learning models **Limitations**: The small size of the dataset and class imbalance may limit the generalisability of the models trained on this dataset **Statistics**: 4530 downloads and 8 citations	
14	Road infrastructure defect dataset kaggle.com/datasets/aniruddhsharma/structural-defects-network-concrete-crack-images	Detecting cracks in the bridge decks, walls, and concrete pavements	56,000 images of cracked and non-cracked surfaces
		**Paper of Origin/Use**: https://www.mdpi.com/2412-3811/7/9/107 **Strength**: Provides a variety of obstructions, such as shadows, surface roughness, scaling, edges, holes, and background debris **Limitations**: Limited to surface cracks only **Statistics**: 2438 downloads and 14 citations	
15	Concrete bridge defects zenodo.org/record/2620293	Concrete bridge surface defect detection	6900 images of the defective concrete surface of 30 unique bridges, including cracks (2507), spallation (1898), efflorescence (833), exposed bars (1507) and corrosion stain (1559)
		**Paper of Origin/Use**: https://arxiv.org/abs/1904.08486 **Strength**: High-resolution images with defects in the context of 30 unique bridges and the use of a multi-stage annotation process resulting in a multilabel dataset with six categories of defects **Limitations**: Varied aspect ratios, scales, and resolutions of defects and even bounding boxes overlap **Statistics**: 27,510 downloads and 80 citations	
16	Road anomaly benchmark github.com/adynathos/road-anomaly-benchmark	Anomalous object detection in autonomous driving and road traffic safety	552 high-definition images of road anomalies and obstacles
		**Paper of Origin/Use**: https://arxiv.org/abs/2104.14812 **Strength**: Provides pixel-level annotations for identifying unseen anomalous objects and hazardous obstacles within diverse scenes **Limitations**: NA **Statistics**: Unknown number of downloads and 34 citations	
17	Pothole detection datasets github.com/ruirangerfan/stereo_pothole_datasets	Pothole detection	220 images of potholes captured using ZED stereo camera
		**Paper of Origin/Use**: https://ieeexplore.ieee.org/abstract/document/8809907 **Strength**: Contains four datasets with disparity maps, designed for pothole detection and published in a reputable journal **Limitations**: NA **Statistics**: Unknown number of downloads and 107 citations	
18	pNEUMA Vision Dataset zenodo.org/record/7426506	On-road anomaly detection	Urban trajectory 35K video frames captured using 18 swarm drones
		**Paper of Origin/Use**: https://www.sciencedirect.com/science/article/pii/S0968090X22003795 **Strength**: Extensive urban trajectory data to investigate traffic phenomena at different scales; provides comprehensive urban trajectory data **Limitations**: Trajectory data are limited to vehicle movement and do not include other factors such as weather, road conditions, or pedestrian behaviour **Statistics**: 417 downloads and 1 citation	

**Table 3 sensors-23-05656-t003:** Prior surveys covering structural defects and anomaly detection.

Research Focus	Research Areas and Applications	Reference
Inspection, defect detection, structural damage, crack detection, ML	Surface defect and damage detection	[8,9,10,13,14,28,62]
Statistical learning, DL, intelligent environments	Anomaly detection, analysis and prediction	[12,16,17,18,20,37,63,64,65,66,67,68,69,70]

**Table 4 sensors-23-05656-t004:** Recent feature extraction-based ML techniques for detecting road anomalies and defects.

Ref.	Detection Origin	Data Acquisition	Algorithm	Evaluation Method	Acc. (%)	Future Implications	Strength	Limitations
Kim, Anagnostopoulos [40]	On-road anomaly	Cameras mounted on a swarm of drones	Butterworth filter, pNEUMA Vision (Dataset 18)	Binary classification, neural network optimisation, precision evaluation	91.8	Improved traffic flow models, enhanced safety analytics, and lane-change detection	Enhanced features, diverse urban traffic use-cases	Bounding box errors, disrupted visibility, tracking failures
Julio-Rodríguez, Rojas-Ruiz [49]	Road surface defects	Vehicle-mounted sensors	KNN, SVM, and RF	Real-world tests on prediction time and classification score	93.20	Improved autonomous driving, energy optimisation, and enhanced vehicle safety	A novel method, real-world tests	Limited feature applicability, idealised scenarios unsuitable for real-time
Ferjani et al., 2022 [50]	Road surface anomalies	Lab simulations and vehicle accelerometer axis data	SVM, decision tree, and MLP	Efficacy of the ML approach using practical, real-world data	94.00	Improved road monitoring, enhanced traffic safety, and reduced accidents	Thorough analysis, practical advice, peer-reviewed, impactful	Feature sensitivity, limited generalisability, domain separation inefficiency
Bustamante et al., 2022 [81]	Road anomalies	GPS, accelerometer, gyroscope, camera	Supervised KNN and ANN	Fog-computing, V2I network using ML algorithms, comparing roughness against a flat reference	95.55	5G-based scalable smart urban mobility, and public spending efficiency	Innovative, data-driven, sustainable urban mobility solution	Data accuracy, privacy and security concerns due to sensors installed inside vehicles
Zhou et al., 2022 [82]	Road Surface Condition	Smartphone camera, accelerator and gyroscope	SVM, KNN, naïve Bayes, decision tree, and RF	Average precision, loss, recall, F1-measure, and accuracy	86.3	Crowdsourcing-based detection system based on motorist feedbacks	High efficiency, low cost, and easy collection of data	Lower accuracy than professional equipment and is affected by shadows, road markings, reflections, and driving habits
Alam et al., 2021 [31]	Debris object detection	Unmanned aerial vehicle (UAV)-mounted cameras	SSD and R-CNN	Mean average precision (mAP) and mean average recall (mAR) scores	88.3	A UAV-based fast and affordable debris detection model for urban planning	The UAV-based method improves road debris clean-up, optimises traffic safety operations	Negative drone distance impact, limited road type scope, detection accuracy affected by environmental factors
Alipour et al., 2020 [83]	Crack detection	Images of diverse road surfaces based on material	ResNet 18-layer, ensemble learning	Accuracy, precision, recall, true negative rate, and F1 score calculated from a confusion matrix	97.8	Construction material independent crack detection model	Robustness of DL methods across various road surface materials	Limited defect types, domain adaptation challenges

**Table 5 sensors-23-05656-t005:** Crack and defect detection methods sorted by publishing date to offer context into the evolution of ARDAD methods.

Taxonomy	Research Focus	Year	Ref.	Acc. (%)	Research Areas and Applications	Implications	Limitations
Traditional ML and statistical methods	Road surface crack, white line, joint detection	2000	[38]	92.8%	Morphology operations for detecting road surface detects to safeguard traffic safety	Automating defect detection	Depend on image quality and require setting parameters
	Lane curvature detection for motorist assistance	2003	[88]	99%	Lane curve and edge detection using a novel image-processing algorithm	Lane departure warning and lateral control system for vehicle control	The proposed algorithm is not effective for road elevations over 2%
	Mobile robot for tunnel crack detection	2007	[89]	NA	Image processing, edge detection, graph search, Dijkstra’s algorithm, expert feedback based	A semi-automated platform for future research in defect detection	Validated in indoor experimental settings with limited application
	Road surface condition recognition	2009	[90]	90%	Road surface condition identification system for motorist safety	Enhancing vehicle active safety features by identifying road surface conditions	Requires extensive vehicle testing for index distribution on road surfaces
	Pavement crack detection	2010	[78]	NA	Pavement edge detection, Canny operator, Mallat wavelet transform, quadratic optimisation	Improving pavement edge detection for faster road repairs to increase road safety	Interference from pavement markings needs further research to counter noise
	Road surface crack detection	2018	[79]	90.87%	ML-based 2D road surface image analysis from the driver’s viewpoint, crack detection, surface defect detection	A platform for cost-efficient, scalable road inspection systems to improve traffic safety	Inefficient in handling varied lighting, shadows, texture, and surface types in image analysis
	Pavement crack detection	2022	[91]	86%	Tile-based image processing method to automate the detection of cracks from 2D and 3D images of pavement and asphalt concrete surface	A platform for an automated pavement distress assessment system, reducing costs and improving the integrity	Limited crack detection, 3D image inconsistencies, false positives, threshold reliance, and width measurement issues
Deep learning	Automatic crack detection on a concrete bridge surface	2011	[35]	90.25%	Image processing, backpropagation neural network, construction safety and management	Automated crack detection system for efficient analysis and visualisation of concrete surface cracks	Performed under similar environmental conditions and needs further evaluation; the accuracy score could be improved
	Road survey for crack detection	2016	[55]	89.65%	ConvNet trained on square image patches, handcrafted feature extraction methods	A platform to build a low-cost, real-time road crack detection system	Misclassification errors in detecting cracks in some of the methods
	Unsupervised multi-scale image fusion	2018	[74]	80.7%	Automated airport runaway inspection using crack detection by multi-scale image fusion	Efficient maintenance of road infrastructures through integration within intelligent autonomous inspection systems	Issues with identical infrastructures, GPS integration and lack of real-time application support
	Road surface cracks and defect detection	2020	[92]	91.99%	Transposed convolution layer, connectivity of pixels, and densely connected layers	An automated solution for detecting cracks in roads and bridges	Poor performance with low-speed cameras; low light conditions affect performance
	Detection of long and complicated pavement cracks	2023	[48]	94.60%	Swin-transformer-based semantic segmentation method with multi-layer perceptron	Improved pavement crack detection, leading to effective maintenance strategies and traffic infrastructure systems	Heavy noise and fallen leaves coupling effect; limited experimental real-world data

**Table 6 sensors-23-05656-t006:** Analysis of various research studies on ARDAD, highlighting their implications for cost-effective rehabilitation decisions and increased traffic safety.

Research Focus	Year	Ref.	Acc. (%)	Research Areas and Application	Implications	Limitations
Automatic crack detection and classification	2009	[98]	94.8% and 95.6%	Entropy, road crack segmentation and dynamic image thresholding	A platform for improved defect detection with cost-effective, objective rehabilitation decision support to increase traffic safety	Potential for improvements in dynamic thresholding accuracy and processing of variance in pixel intensity
Image processing for pothole detecting	2015	[99]	77.9%	Pothole detection using simple real-world images, Canny filter and contour detection	A device for vehicles that detect potholes, alerts drivers, and log pothole locations for road maintenance agencies	Limited detection range, potential for absorption of potholes into outer borders, and inability to detect potholes with no visible edges
Crack detection on two-dimensional pavement images	2016	[100]	83%	Crack detection, minimal path, Dijkstra algorithm, road surface condition analysis	An unsupervised learning algorithm for effective assessments in road in road maintenance	Potential bias due to the use of the same Dijkstra algorithm and high computation time requiring optimisation for faster processing
Road defect detection	2016	[101]	54% to 91%	Road defect detection, image processing, computer vision	A real-time road defect detection system for timely road repair and traffic safety	Issues with road colour affecting accuracy rates, real-time constraints, difficulty detecting thin cracks
Road condition detection system	2018	[43]	87%	Road Weather Information System (RWiS), Intelligent Transportation System (ITS)	Improving traffic safety by enabling autonomous cars to avoid road anomalies and control speed based on road condition	Issues with background noise filtering resulting in object shadows being detected as cracks
Automatic road crack segmentation	2020	[77]	99.11%	Morphological filter dynamic thresholding, entropy thresholding	A high-performance model for crack detection	The model presented does not address characterising crack severity

**Table 7 sensors-23-05656-t007:** Various deep-learning-based research in yearly ascending order for reference to give context into the evolution of ARDAD methods.

Research Focus	Year	Acc. (%)	Ref.	Research Areas and Application	Implications	Limitations
Wildlife-vehicle collision analysis and hotspot prediction	2006	High *p*-value 0.463	[1]	Linear nearest neighbour analysis, Ripley’s K analysis, visual analysis	Identifying hotspots to aid transportation agencies to mitigate wildlife-vehicle collision	Limited collision hotspot data for the initial method improvement
Detection and counting of potholes	2016	83.18%	[80]	K-means clustering-based segmentation, image processing, edge detection, identification, segmentation	A standalone application for pothole detection using hybrid classifiers	The study provides an analysis but no solution for pothole detection
Detecting road hazards to help self-driving vehicles	2016	TPR of 63%	[29]	DBSCAN, image processing using stereo-based baseline methods, clustering	Improving self-driving vehicles to detect small road hazards and help decrease accidents caused by road debris could be reduced	Limited to stereo-based methods and specific datasets, missing real-world scenarios
Pothole and hump detection	2018	NA	[76]	Internet of Things (IoT)-based road-monitoring system, honeybee optimisation (HBO), cloud-based real-time image processing	Improving traffic safety via timely alerts for motorists and facilitation of road maintenance	Study only tested two-speed scenarios (40 km and 60 km); real-world implementation not assessed
Road anomaly detection	2020	F1 of 66.7% to 92.1%	[94]	Threshold detection, sliding window, KNN dynamic time warping	Large-scale data integration to create city-wide anomaly maps	Sensitive to noise, does not fully represent diverse road conditions
Video surveillance, anomaly detection	2011	F1 of 55%	[39]	Particle-based tracking, a cascade of HMM and HDP-HMM models	A solution working on less structured CCTV footage, such as videos of metro systems	Manual parameter setting, inability to distinguish pedestrian and vehicle activities
Real-world surveillance video anomaly detection	2018	Approx. 95% (Accidents)	[105]	Multiple instance learning (MIL), deep MIL ranking model, temporal segmentation	Improved anomaly detection by reduced reliance on manual annotations and enhanced real-world anomalous activity recognition capabilities	Potential false positives, reliance on weakly labelled data, computational complexity, and dataset diversity constraints
Road surface monitoring	2018	97%	[54]	Support vector machine, hidden Markov model (HMM) and residual network (ResNet)	Enhanced road monitoring and maintenance through smartphone-based data collection and analysis	Limited dataset, manual labelling, vehicle and smartphone variability, lack of road roughness estimation
Anomalies detection for autonomous vehicles	2021	99.21%	[106]	Image processing, AI-based edge computing for vehicular ad hoc network (VANET)	The scalable edge computing AI-based framework could improve traffic safety and autonomous driving by providing real-time road information	Limited dataset collected from online sources and the need to incorporate more types of road anomalies

**Table 8 sensors-23-05656-t008:** Soft-computing approaches: traditional ML vs. DL systems.

	Traditional ML	Deep Learning
Strength	(i) Learning is possible even with a small dataset (ii) It does not require high computational power or resources (iii) Working architecture is relatively easy to interpret (iv) Works well with structured data	(i) Works well with unstructured data (e.g., audio, video and multi-time series) (ii) Possibility for automated feature extraction from data (iii) Can provide end-to-end solutions
Weakness	(i) Domain expertise may be required in handcrafting of feature extraction algorithms (ii) Problems often need to be broken down to find solutions for the subproblems and the outcomes produced may need to be combined	(i) Requirement of large datasets in learning (ii) The working architecture is complex containing typically a large number of parameters (iii) Training data set size and computing resources required for model training

**Table 9 sensors-23-05656-t009:** Literature since 2018 on ARDAD systems based on or future directions towards 3D imaging, laser or LiDAR sensors.

Research Focus	Year	Ref.	Acc. (%)	Research Area and Application	Implications	Limitations
Pavement surface reconstruction for crack recognition	2018	[116]	78.27%	Microsoft Kinect fusion for crack detection, surface reconstruction for pavement serviceability analysis	Upgraded Kinect hardware and expanded data sources for enhanced pavement assessments and traffic safety	Limited hardware capabilities and Kinect field-of-view constraints hinder capabilities
Road crack and pothole detection	2018	[21]	98.93%	Using texture-based features to differentiate between crack surfaces and sound roads	Enhanced road monitoring and maintenance, reduced accidents, and improved navigation for autonomous vehicles	Inefficient restoration patches detection, issues with shadow, occlusions, and camera resolution limitations
Road anomaly detection	2018	[117]	82.51%	Principal component analysis, Fi-Ware, data mining, collaborative mobile sensing	Improve data acquisition standardisation, sensor diversity, and merging long/short bump classes to enhance real-world performance	Decreased performance in real-world conditions, data standardisation and complexity reduction not effective
Speed bump detection	2019	[118]	97.14%	Self-driving cars, artificial vision, GPS tracking	Real-time road surface monitoring, smart route optimisation, reduced fuel consumption, and continuous updates of road quality	Model uses both accelerometer and gyro data; improved performance with only one source not yet achieved
Anomaly detection	2019	[17]	NA	Smart objects, intelligent transportation systems, industrial systems	Prediction/prevention and exploring data fusion techniques	Limited data access, focus on normal behaviour, high-dimensionality issues
Asphalt pavement crack classification	2019	[86]	87.50%	Asphalt pavement, crack classification, image processing, steerable filters	Image processing methods to assess crack properties, including depth and severity	Limited crack types, small image dataset, unexplored crack properties
Road crack detection	2019	[84]	98.70%	Deep learning and adaptive image segmentation	A deep neural network trained to segment positive images into semantically meaningful regions, i.e., cracks and road surface	Difficulty in properly segmenting colour images with a large number of noisy pixels
Low-cost pavement condition health monitoring	2020	[119]	93.55%	Automated detection of road pavement distresses, low-cost DL technologies	Platform for an integrated approach towards optimising urban pavement management systems	Region-specific model with manual data collection dependence, no detailed quantitative assessments
Road damage detection	2020	[96]	56.5%	EfficientDet model for crack and object detection	Improving results by setting ground rules for annotating and expanding datasets by installing cameras with optimal orientation	Limitations of the study include false positive and negative detections, misclassifications between diagonal crack classes
Road surface monitoring and pothole detection	2020	[22]	85% and 93%	Deep learning, road surface monitoring, pothole detection, crowd sensing	Adaptive method to analyse the additional type of road surfaces and apply end-user driving profiles	Limited road surfaces analysed, controlled scenarios in related works, incomplete automatic threshold adjustments
Asphalt pavement crack detection	2020	[71]	70%	YOLOV3-based asphalt pavement crack and pothole detection	Improving road maintenance efficiency and reduce infrastructure costs using AI-driven analysis	Limited by data geographical scope and weather variations, human judgment discrepancy affecting model performance
Pavement distress and health monitoring	2020	[103]	89.14%, 97.66%	Road pavement distresses detection, minimal annotations learning	Suggested trends and future work include exploring activation functions, selective layer freezing, transfer learning and different CNN architectures	Limited real-world anomalous samples and potential impact of activation function choice in transfer learning
Modern Pothole detection technique	2020	[104]	NA	TensorFlow API, transfer learning, road inspection automation	Improved CNN architectures, GPS-enabled systems and Android apps, deployment on Raspberry Pi or Android devices	Limited detection in varying conditions, computational complexity, generalisability issues and integration challenges
Object and anomaly detection	2020	[111]	59.11%	Amazon Rekognition, Azure Cognitive Services and Google Vision	Choosing cloud platforms and edge devices for IoT applications based on performance and cost trade-offs for improved traffic safety	Limited number of tested platforms and the lack of real-world deployment
Crack detection of concrete pavement	2020	[120]	90.1%	Crack detection, cross-entropy loss function, VGG16 network, crack classification	Integration of algorithms with other technologies, such as drones and robots for automated inspection and maintenance	Lack of external validity in terms of deploying the proposed algorithm in real-world scenarios
Road damage detection	2020	[42]	66%	Ensemble learning, object detection, urban street analysis	An automated solution for road damage detection and classification using image analysis for smart city applications	Lack of diversity in the training dataset and limited evaluation of real-world scenarios
Road damage detection	2020	[112]	63.58%	Image classification, object detection, and ensemble models	Improving road safety and developing better road damage detection systems using smartphone and vehicle-mounted cameras	The need for high-quality data and the impact of input image size on detection performance
Predicting the accuracy of asphalt concrete pavement	2021	[51]	NA	AdaBoost regression, International Roughness Index (IRI), Mechanistic-Empirical Pavement Design Guide (MEPDG)	Improving pavement design, understanding of influencing factors (including reported variables analysis) and optimising costs of road maintenance	Limitations include data bias, overfitting, lack of interpretability, and generalisation to new contexts

## Data Availability

Not applicable.

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
