# Peer review of "Automated Road Defect and Anomaly Detection for Traffic Safety: A Systematic Review"

_sensors, 2023, doi:10.3390/s23125656_

Round 1

Reviewer 1 Report

This paper presented a detailed review automated road defect and anomaly detection, where they divided the area into logical categories, and reviewed papers published between 2000-2023.  First of all, as a review paper, it was reasonably comprehensive, and quite well written, with a logical classification system, and reasonably good coverage.  However, there are a number of areas that can be improved.  

General comments

----------------

Firstly, with regard to writing, there are a couple small examples of inconsistency, for example, we have AR-DAD and ARDAD.  We also have mixed up tenses throughout the paper, where researchers "propose" something and then later in the same paragraph, something is "proposed".  Make sure your tenses are consistent.

With regard to content, the key issue I have is that I don't think the review is critical enough.  For me, it felt like rather than reviewing, the authors just list things.  If its a review, I expect more detail.  For example, in the datasets, they list a number of public domain datasets and a few details, but I don't really get a sense of strengths and weaknesses.  Which datasets are more widely used?  What are the limitations with them?  How versatile are they?  What would be recommended, and what would be best avoided?  The same applies throughout the paper.  I think there needs to be more critical review.

Specific Details

----------------

1.  Abstract, you use AR-DAD and ARDAD interchangeably, also se-lected (line 17), and I'm not sure if "artefact" is the correct word to be using.

2.  Introduction, line 119.  I'm not sure that "a team of experts" is true.

3.  Section 2, I don't think figures 4 and table 1 are needed.  I don't know what they bring, when it could be summarised fairly quickly.  I would be tempted to remove pretty much all this section.  Just list how many papers you reviewed.  You only need a couple of sentences.  Most of the text is not required.

4.  Figure 5 contains a graph of number of papers published.  Later, they discuss how they feel the numbers look low because of their selection criteria, and I agree.  I think its a bit misleading in terms how its grouped, and if this data is to be presented, it should be a more comprehensive list, and a line chart sorted by year would be a better format.

Datasets

5.  Datasets should be a section of its own, not a subsection of the review questions.  I discussed it above, but more detail is needed.

6.  "Care has been taken to select datasets that do not require a login and are behind paywall barriers".  Why?  If you're producing a review, and some papers use datasets that are proprietary or require registering, then they should be listed?

7.  Table 2 is reasonably good, but I would also like to know a bit more information.  Which of these datasets are out of date?  Which are benchmarks and widely used?  How many citations do they have?  Maybe link to a couple papers that use each of these datasets?  

8.  Relating to the above, perhaps as well as listing the datasets in the table, you should provide a more detailed written critical review?

Lit Review

9. You make it clear that you are only focusing on 2020 - 2023, but personally I would have liked to see a small historical review to provide context.

10.  Lines 2662 - 265 (and other places).  If you are listing examples, then cite them!    

11.  In line 275, Li et all "employed" three ..., where as in line 283, Chatterjee and Saeedfar "present" an improved...  You should be consistent.  I'm not going to identify all cases of this, so it is your responsibility to check through the paper.

12.  THis section is an example of non-critical review.  They list a lot of examples, but at the end, I have no idea what the state-of-the-art best results are, and what techniques are out of date.  I don't know what datasets these papers listed in table 4 are, and what the strengths and limitations of the key papers discussed are.  I would like more detail.  For example, in table 4, why not link the datasets back to the dataset section?

13.  Same suggestion applies throughout the paper.  As a basic review of listing what people ahve done, its fine, but as a criticial review, I feel it needs a little more.

14.  Line 337, I don't know if I agree with the premise that unsupervised learning models may be superior, purely because of no human input, but that's just an opinion. However, "and are, therefore may be superior" needs to be rewritten.

15.  Same comment as #12 for all other tables.  

16.  Line 433 shows an example of the steps and the results, but in comparison, lines 421 to 429 just list several techniques without any real details.

17.  I'm not sure what Figure 6 brings to the paper, apart from being a nice graphic.  I'd appreciate the thoughts of the authors.

18.  I'd like to see more of a summary of the different sections.  Based on your expertise, what are the most promising techniques to explore further?

Overall

-------

Overall, I think its a reasonably well written and easy to read review, but I do feel that with a little more work, it could become a very good review.  I would encourage the authors not to rush the changes, and to take the time to do a detailed revision.

Author Response

Reviewer 1 Colour code 

Thank you for your valuable feedback on our manuscript. We have carefully addressed the raised concerns and incorporated the suggested revisions. We have introduced colour formatting to indicate specific modifications based on your feedback to improve readability and highlight the updates. In summary, we have rectified formatting inconsistencies, justified using the term "artefact" in the abstract, and revised the phrase "a team of experts" to better convey our intended meaning. We have provided clarification on the importance of Figure 4, Table 1, and Figure 5 and restructured the manuscript to include a dedicated section for datasets.

Additionally, we have expanded on the details and critical reviews of the datasets and techniques discussed. The Conclusion section has been updated to reflect these major changes. Overall, we have enhanced the critical analysis, expanded discussions, and provided more comprehensive summaries throughout the paper. Please refer to the table below for a comprehensive overview of our point-by-point responses to each reviewer's feedback.

  • 1. Abstract, you use AR-DAD and ARDAD interchangeably, also se-lected (line 17), and I'm not sure if "artefact" is the correct word to be using.)

Thank you for your valuable feedback. The errors appeared due to a mismatched MDPI template with our local version of Microsoft Word, resulting in unintentional formatting inconsistencies. We have now examined the manuscript for the correct usage of abbreviations and uniform tenses throughout the paper. 

Regarding the use of the term "artefact" in the abstract, we used this term to refer to the various elements and resources discovered in our systematic reviews, such as datasets, research trends, and technology advancements. We believe that "artefact" is suitable in this context, as it represents the tangible outcomes of our detailed study of the ARDAD and CV domain. Here is a snippet of the changes made in the abstract section: “As enabling factors, computer vision (CV) combined with sensor technology, have made progress in applications intended to mitigate high rates of fatalities and the costs of traffic-related injuries. Although past surveys and applications of CV have focused on subareas of road hazards, there has yet to be one comprehensive and evidence-based systematic review to investigate CV applications for Automated Road Defect and Anomaly Detection (ARDAD).

  • 2.  Introduction, line 119.  I'm not sure that "a team of experts" is true.

We understand your point of view and agree that the phrase "a team of experts" could be misleading. However, our intention was to highlight the reliance of traditional machine learning models on domain-specific knowledge and expert input, particularly in the context of data pre-processing and feature extraction. While it's not always necessary to have a "team" of experts, we believe that the input of at least one expert can greatly enhance the model's performance and accuracy.

To give an example, during the development of my master's thesis, I initially misclassified a false positive as a true positive. It was only after consulting with a domain expert of the New Zealand Transport Agency that I was able to correct this error. This interaction not only improved the accuracy of my model but also led to the creation of a third class that was not strictly a "true positive" but required attention nonetheless.

Based on this personal experience, we remain convinced of the importance of expert involvement in the development of machine learning models. However, we appreciate the ambiguity of the phrase "a team of experts" and have revised the text to “Traditional ML models often rely on domain knowledge or domain experts to design or improve data pre-processing and feature extraction algorithms” on lines 118 and 119 to better reflect our intended meaning. Thank you once again for your insightful feedback. 

  • 3.  Section 2, I don't think figures 4 and table 1 are needed.  I don't know what they bring, when it could be summarised fairly quickly.  I would be tempted to remove pretty much all this section.  Just list how many papers you reviewed.  You only need a couple of sentences.  Most of the text is not required.

We would like to emphasise the importance of Figure 4 and Table 1 in our manuscript. These elements provide insight into our modified PRISMA checklist specifically tailored for ARDAD research. Our modifications represent an original contribution to the field of computer vision and ARDAD; as such, a tailored PRISMA checklist has not been presented before.

Including the modified PRISMA checklist in Section 2 is necessary to ensure repeatability and demonstrate its applicability to the unique aspects of ARDAD research, such as the use of machine learning and data mining techniques, the diversity of data sources, and the rapidly evolving nature of the field. We believe that these modifications will improve the quality and transparency of systematic reviews in ARDAD research and facilitate the synthesis of evidence for future research and practice. 

  • 4.  Figure 5 contains a graph of number of papers published.  Later, they discuss how they feel the numbers look low because of their selection criteria, and I agree.  I think its a bit misleading in terms how its grouped, and if this data is to be presented, it should be a more comprehensive list, and a line chart sorted by year would be a better format.

Thank you for your valuable feedback. We have revised Figure 5 to a scatter plot to better convey the trends and changed the caption (lines 209-211) to "Figure 5 scatter-plot distribution shows the relationship of the number of articles reviewed on the various types of anomalies from January 2000 to May 2023 (i.e., date of publication).

  • 5.  Datasets should be a section of its own, not a subsection of the review questions.  I discussed it above, but more detail is needed. 

We are grateful for your valuable feedback concerning the organisation and depth of our dataset review and the selection criteria for the datasets included. In response to your suggestion, we have restructured our manuscript to make datasets their own dedicated section, separate from the review questions. This change allows for a more comprehensive and focused discussion of the various datasets.

  • 6.  "Care has been taken to select datasets that do not require a login and are behind paywall barriers".  Why?  If you're producing a review, and some papers use datasets that are proprietary or require registering, then they should be listed?

We initially considered including proprietary datasets or those requiring registration but discovered that numerous relevant and helpful open-access datasets were already available. We wanted to cater to researchers from all backgrounds and not discriminate in favour of the elite members of the community who may have easier access to restricted resources.

During our initial selection process, we encountered some dataset links that led to unsafe web pages. In the interest of our readers and the wider research community, we prioritised providing one-click access to free-of-cost, safe, and relevant datasets. This approach ensures a more inclusive and secure experience for researchers seeking to utilise the datasets in their work. Here is the additional paragraph in response to the question: “The motivation for assembling this diverse dataset is to provide unrestricted access without login requirements or paywall barriers. This section reports a selection of single-point, one-click access to frequently downloaded open-access datasets for the research community (Table 2). Our goal is to promote inclusivity and remove possible discrimination, ensuring that researchers from all backgrounds can contribute to and benefit from the advancements in the field of ARDAD. To verify unrestricted, all-inclusive access and to promote privacy, we have tested the dataset access to ensure that all data are freely accessible without needing a login or being restricted by paywalls.

  • 7.  Table 2 is reasonably good, but I would also like to know a bit more information.  Which of these datasets are out of date?  Which are benchmarks and widely used?  How many citations do they have?  Maybe link to a couple papers that use each of these datasets? 

we understand the importance of providing a balanced and thorough evaluation. we have expanded our review to provide more detailed information on the strengths and weaknesses of each dataset. We now offer insights into their evaluation criteria and highlight the advantages and limitations of their usage in the field.

  • 8.  Relating to the above, perhaps as well as listing the datasets in the table, you should provide a more detailed written critical review?

Thank’s again for your valuable feedback; we have significantly expanded the dataset table and incorporated all the critical information pertaining to each dataset. Including the critical review directly within the table under each dataset link offers the reader an easy-to-follow and comprehensible format instead of presenting it as a paragraph of text. This approach ensures that the information is both accessible and well-organized for quick reference and comparison.

  • 9. You make it clear that you are only focusing on 2020 - 2023, but personally, I would have liked to see a small historical review to provide context.

We have added a paragraph (lines 130 to 137) to our introduction that captures the evolution of ARDAD methodologies and technology from 2000 to 2023, providing the historical context you suggested ”The motivation for this systematic review lies in the understanding that road defects and anomalies significantly impact traffic safety and the overall economy. The studies from 2000 to 2023 in this systematic review are selected to capture the evolution of ARDAD methods and technology over the past two decades. The selected time frame covers crucial developments, including an elementary mathematical morphological method at the turn of the millennium [38], automated anomaly detection a decade later [39], and, ultimately, sophisticated surveillance techniques employing UAV swarms by 2023 [40]. ”.

  • 10.  Lines 262 - 265 (and other places).  If you are listing examples, then cite them!  

Thank you for pointing out the need for citations in the listed examples. We have carefully reviewed the manuscript and have now provided the appropriate citations where required. From 297-300: “Image processing based on traditional methods (statistical and classical ML) has been used to analyse road sections' images to detect defects [10]. Examples of applied methods in image processing include logical and linear regression [55], naïve Bayes [66], support vector machine (SVM) [52], random forest (RF) [56] and more.

  • 11.  In line 275, Li et all "employed" three ..., where as in line 283, Chatterjee and Saeedfar "present" an improved...  You should be consistent.  I'm not going to identify all cases of this, so it is your responsibility to check through the paper.

Thank you for your valuable feedback. We have carefully revised the manuscript and have now uniformly changed the wording to "proposed" throughout the paper. We believe that this change not only addresses your concerns but also aligns better with the terminology commonly used by the majority of authors in the field.

  • 12.  THis section is an example of non-critical review.  They list a lot of examples, but at the end, I have no idea what the state-of-the-art best results are, and what techniques are out of date.  I don't know what datasets these papers listed in table 4 are, and what the strengths and limitations of the key papers discussed are.  I would like more detail.  For example, in table 4, why not link the datasets back to the dataset section?

Thank you for your valuable feedback regarding the lack of critical analysis and detail in Table 4. In the revised manuscript, we have updated Table 4 by adding multiple new columns to incorporate evaluation methods, future implications, strengths, and limitations of the key papers discussed.

However, we would like to clarify that we have not provided backlinks to datasets for all papers because not all the papers listed in Table 4 grant open access to their datasets. Wherever applicable, we have cross-referenced the papers to Table 2 (procured datasets). In the spirit of inclusivity and promoting open-access resources, we have refrained from listing datasets that are not publicly accessible.

  • 13.  Same suggestion applies throughout the paper.  As a basic review of listing what people ahve done, its fine, but as a criticial review, I feel it needs a little more.

Thank you, in response to your suggestion, we have taken steps to enhance the critical analysis in our manuscript. We have expanded our discussions to include more detailed evaluations of the strengths, limitations, and implications of the reviewed algorithms, datasets, and techniques. We have also included relevant performance metrics, such as accuracy, F1 scores, and other measures, to support our evaluations. 

Furthermore, all tables, including Table 4, have been updated to include additional columns on future implications, accuracy, and limitations. We have incorporated our expertise and insights to provide a critical perspective on the state-of-the-art techniques and their future implications. By discussing the potential advancements, promising directions, and areas for improvement, we aim to contribute to the field by offering valuable insights and recommendations for future research endeavours.

  • 14.  Line 337, I don't know if I agree with the premise that unsupervised learning models may be superior, purely because of no human input, but that's just an opinion. However, "and are, therefore may be superior" needs to be rewritten.

We have carefully considered your comment and revised the relevant section in our manuscript. The updated text now highlights the potential advantages of unsupervised learning models in detecting on-road anomalies without making a definitive claim of superiority. We believe this revision provides a more balanced perspective on the topic (lines 370 to 374): In detecting on-road anomalies, unsupervised learning models hold potential advantages as they do not rely on labelled data for sample classification, unlike supervised learning models, which depend on subjective human input [92, 93]. As a result, the output of unsupervised learning models is not predetermined, allowing computers to independently discern anomalies in the data through classification processes [94].

  • 15.  Same comment as #12 for all other tables. 

Thank you for your comments. We have made the necessary updates to all the  tables, including Table 4, to incorporate accuracy percentage, future implications, application areas, and limitations. This will provide a more comprehensive and critical perspective for our readers.

  • 16.  Line 433 shows an example of the steps and the results, but in comparison, lines 421 to 429 just list several techniques without any real details.

In response to your concerns, we have revisited the mentioned section of the manuscript (lines 477 to 484) and have made necessary revisions to provide more in-depth information and relevant facts about the techniques discussed: The twice-threshold segmentation method demonstrates higher accuracy of up to 98% in detecting cracks in runway images containing road markings, outperforming traditional threshold segmentation algorithms like Otsu (40%) while maintaining adaptability for various applications [110]. Amhaz and Chambon [99] proposed the Minimum Path Selection (MPS) algorithm for crack detection with a Dice Similarity Coefficient (DSC) of 0.77 on 2D pavement images. However, further advancements in computation time and adaptability to 3D imaging systems are necessary for broader applications.

  • 17.  I'm not sure what Figure 6 brings to the paper, apart from being a nice graphic.  I'd appreciate the thoughts of the authors.

We appreciate your comments regarding Figure 6. Therefore, we have added to the caption the knowledge this figure offers to the readers as it demonstrates a snapshot of the current machine-learning trends used in road defect detection. Furthermore, k, we have included further explanations of Figure 6 and its significance in the revised manuscript, lines 577-588. We have also provided brief summary at the end of the sections: “Figure 6 depicts the volume of literature reviewed based on detection origins divided by the ML methods' taxonomy. Deep learning (34%) is the most popular method, followed by traditional ML (26%) and ensemble learning (26%). In comparison, 3D Image-based techniques (14%) are the least represented by the reviewed ARDAD systems. Considering the timeframe of the literature reviewed, recent advancements in ML techniques may impact the overall taxonomy distribution and the road defect detection landscape. The growing popularity of DL approaches is likely due to their ability to process large datasets and automatically extract relevant features. However, Traditional ML and Ensemble Learning methods are still widely used across ARDAD systems. However, in discussing the benefits and drawbacks of each taxonomy, it is essential to acknowledge the gaps in the literature and encourage further research to explore underrepresented ML methods or road defect types.

  • 18.  I'd like to see more of a summary of the different sections.  Based on your expertise, what are the most promising techniques to explore further?

The review highlights the significance of road defects and anomalies, which have serious implications in terms of cost and safety. The paper discusses the potential of automating the road infrastructure supervision process and presents a selection of resources, including open-access datasets, technology trends, and research directions, that can help accelerate progress in the field of Automated Road Anomaly and Defect Detection (ARDAD) and Computer Vision (CV). The review covers different types of road defects and anomalies, such as potholes, cracks, debonding, stripping, ravelling, bleeding, shrinkage, and swelling, and presents several detection methods, including image processing-based, traditional machine learning, deep learning, and feature extraction-based machine learning methods.

Based on our expertise, we recommend exploring ensemble methods that combine multiple techniques for detecting road defects and anomalies, which have shown promising results. Furthermore, incorporating advanced sensors such as LiDAR and multimodal data fusion techniques can improve the accuracy of detection. We also recommend leveraging crowdsourcing methods for data collection and system validation, as this can improve accessibility and reduce costs.

Reviewer 2 Report

This paper reviews the technology of Automatic Road Defect and Anomaly detection (AR-DAD) for traffic safety since 2000, mainly based on road image and video processing algorithms. This paper reviews the main literature on AR-DAD since 2000, and summarizes various major defect detection algorithms, including traditional image processing algorithms, machine learning algorithms, deep learning algorithms and computer vision algorithms. The purpose of this paper is to help people improve traffic safety. The paper has the following problems:

1. The description of various road defect and anomaly detection algorithms is not deep enough, and some even mention the name of the algorithm without summarizing the characteristics of various algorithms. The comparison between traditional algorithm and deep learning algorithm is only given, without giving the comparison of similar algorithms. Therefore, the comparison results given in the paper are not convincing.

2. The algorithm classification and order in the paper review are chaotic. When describing traditional algorithms, there is a lot of machine learning content mixed in. When describing deep learning algorithms, traditional algorithms are also mentioned, and no summary is provided for each reviewed algorithm;

3.      The main purpose of the paper is to improve traffic safety, but there is too little literature and technical review on improving traffic safety.

Conclusion: major revision is required.

Author Response

Reviewer 2 Colour code 

We appreciate your valuable feedback, which has helped refine our manuscript and enhance its quality. We acknowledge the comment regarding the depth of our description of various algorithms and agree that further elaboration would strengthen our analysis. As a result, we have expanded our discussion on each algorithm, providing comprehensive summaries of their characteristics, strengths, limitations, and performance metrics. We have also included comparisons between similar algorithms, presenting accuracy, F1 scores, and other relevant measures for a more robust evaluation.

Regarding the classification and order of algorithms, we understand the concern raised. While our focus was on defect types and anomaly detection in automated road defects and anomaly detection (ARDAD) systems, we have made significant revisions to reorganise and reclassify algorithms, ensuring a clearer distinction between traditional machine learning (ML) and deep learning (DL) algorithms. Each algorithm now has a concise summary highlighting its characteristics and contributions.

We recognise the importance of highlighting the relevance of our review to traffic safety. Our review is centred around this aspect, analysing ARDAD systems and their implications for traffic safety. We have made efforts to further emphasise the traffic safety applications addressed by individual algorithms and their future implications. We believe these additions provide a clearer connection to traffic safety throughout the manuscript.

  • 1. The description of various road defect and anomaly detection algorithms is not deep enough, and some even mention the name of the algorithm without summarizing the characteristics of various algorithms. The comparison between traditional algorithm and deep learning algorithm is only given, without giving the comparison of similar algorithms. Therefore, the comparison results given in the paper are not convincing.

Thank you, and we appreciate your valuable feedback, which has helped us refine our manuscript and enhance its overall quality.

We acknowledge your comment regarding the depth of our description of various algorithms. Upon careful consideration, we agree that further elaboration on the characteristics of each algorithm would strengthen our analysis. As a result, we have expanded our discussion on each algorithm, providing a more comprehensive summary of its key features, strengths, limitations, and performance metrics. In addition, we have included comparisons between similar algorithms, presenting accuracy, F1 scores, and other relevant measures to provide a more robust evaluation.

We also appreciate your suggestion to explicitly address the future implications of the different algorithms. In response to your feedback, we have revised the manuscript to explicitly discuss each algorithm's potential applications and advancements. By doing so, we aim to provide a clearer understanding of their long-term impact in the field of road defect detection.

  • 2. The algorithm classification and order in the paper review are chaotic. When describing traditional algorithms, there is a lot of machine learning content mixed in. When describing deep learning algorithms, traditional algorithms are also mentioned, and no summary is provided for each reviewed algorithm;

We acknowledge your observation regarding the classification and order of algorithms in our review. While we agree that the initial flow may not have strictly followed a separation between machine learning (ML) and deep learning (DL) algorithms, we would like to highlight that our primary focus was on the types of defects and anomalies in automated road defects and anomaly detection (ARDAD) systems. We aimed to emphasise the detection types, such as visco-plasticity, bleeding, crack types, and various anomaly types, which influenced the overall flow of our review. We

That being said, we understand the importance of providing a clear distinction between traditional ML algorithms and DL algorithms. In response to your feedback, we have made significant revisions to the manuscript, reorganising and reclassifying certain algorithms to address this concern. We have also ensured that each algorithm is accompanied by a concise summary highlighting its characteristics and contributions.

Our analysis suggests that in the field of ARDAD systems, the distinction between traditional ML and DL algorithms is not always straightforward or necessary, as the detection types and anomaly characteristics often take precedence. Nonetheless, we have made necessary adjustments to improve the structure and flow of the review based on your valuable feedback.

  • 3.      The main purpose of the paper is to improve traffic safety, but there is too little literature and technical review on improving traffic safety.

While we understand the importance of explicitly highlighting the relevance of our review to traffic safety, we would like to clarify that our entire review is centred around this fundamental aspect. Our focus on various defect types and anomaly detection methods directly relates to their impact on traffic safety. We intend to comprehensively analyse ARDAD systems and their implications for traffic safety. We value your feedback and have made efforts to highlight further the traffic safety applications addressed by individual algorithms and their future implications. We hope that these additions address your concerns and provide a clearer connection to traffic safety throughout the manuscript. 

After the expansion and modifications, we have also updated the Conclusion section; I am putting it out here for your consideration because it has been modified mostly based on your feedback: 

“Motivated by the need to accelerate technological advancements that can improve traffic safety and reduce incidents, this systematic review analyses the literature on automated road defects and anomaly detection (ARDAD) systems from 2000-2023. As a result, the systematic review covers peer-reviewed articles (N=116) associated with types of roadside anomalies and defects that are jointly intended to help prevent loss of lives, injuries and infrastructure damage, ensuring on-road and structural integrity.

In the context of augmenting on-road surveillance for ease of maintenance, such as structural damage detection and hazard prevention via predictive monitoring, the review summarises the ARDAD methods, including the achieved performance using traditional ML, and DL, combined with sensor technology. Notably, it quantifies the achieved performance of these methods, providing insights into their effectiveness. Additionally, the review provides a taxonomy of ARDAD methods and descriptions, including a list of frequently downloaded open-access on-road anomalies and defect image datasets (D=18), facilitating future research and benchmarking.

Considering the current publication trends, the advancements in video technology, availability of sensors and computing resources in general, there is an exponential growth in ARDAD research publications from 2000 to the present day. As anomaly detection intersects with automatic road traffic surveillance, this survey can also be a valuable resource for interested researchers working on related contexts.

As a consequence of the impact of the global pandemic and lockdowns in the period from 2020 to 2022, there was less traffic, and opportunities for new data collections compared to the previous years. The exponentially growing trend in the number of research publications during the period from 2015 to 2020 could be explained by earlier data collections, prior to the global pandemic (Figure 5). In the authors’ view, the growing trend surrounding ARDAD technologies and research is yet likely to reach its peak, aligning itself with the early stage of “Gartner’s technology adoption hype cycle framework, (Step 1)”. As such, future work on ARDAD technologies is likely to consider Gartner’s framework for better understanding of a current project position on the hype cycle, to project the adaptation and maturity levels (of ARDAD technologies), to identify practical aspects of technology transfer such as self-driving vehicles and to identify possible impact on society.

Considering the state-of-the-art ARDAD methods, we conclude that the latest IoT, 5G and 6G communication technologies, swarm drones, satellite imagery, cloud computing and GPS, have the potential for near-future research and further expansion of related research contexts. The benefits of ARDAD methods to humanity include the utilisation and advancements of AI, CV, and semi and self-learning techniques to support intelligent vehicles, urban planning, intelligent transportation systems, connected or self-driving vehicles, improved road surveillance, reduced road maintenance costs, and increased traffic safety.

In order to enhance future research in the field of ARDAD systems, there is a crucial need for more comprehensive performance/meta-analyses that can evaluate the efficacy and efficiency of various ARDAD methods in real-world settings. While not a full meta-analysis, our systematic review provides a strong foundation serving as a platform for future research. This potential conversion would enable a quantitative data synthesis, further advancing our understanding of ARDAD technologies and facilitating evidence-based decision-making. Additionally, quantifying the societal and stakeholder impacts resulting from the implementation of ARDAD systems would offer valuable insights for policymakers and industry professionals.

Overall, this systematic review is a significant milestone in ARDAD systems, bridging a crucial research gap with its comprehensive analysis of traffic hazards ranging from urban cities to the wild hinterlands. Our commitment to inclusivity is evident in examining often-overlooked road hazards like avalanches or cattle on the road, showcasing our genuine belief in uncovering hidden knowledge from future data or previously unseen or untested datasets. This systematic review establishes a foundation for future research endeavours in ARDAD systems and highlights the potential of emerging technologies to drive advancements in traffic safety and road maintenance. Our research findings inspire optimism based on emerging technologies' potential to facilitate advancements aimed at improving safety and saving lives and making a positive impact on global society.

Round 2

Reviewer 1 Report

The authors have addressed most of my comments and have made big changes.  I think its a much improved paper.

The only area I have an issue with is that not all datasets are free, and I understand the authors argument that they were looking for free, accessible, one click datasets, but then some of the papers they cite use non accessible corpora (I believe).  My personal preference would be to add a small subsection which lists some of the other datasets that key papers have used, while explaining that access is limited.  However, I will leave that to be an editorial decision.

There are a few small grammatical errors, so check the paper over carefully before finalising.

Author Response

We appreciate and thank you for taking out time to provide the second review. We have included a new subsection in our revised manuscript. This subsection provides a list of datasets used in some of the studies included in our systematic review, acknowledging their limited accessibility: “Apart from the datasets provided in Table 2, the systematic review inspected datasets used by the studies which are not open-access. This leads to identifying datasets available upon request or needing a paid subscription. For instance, the research on pavement crack detection [48] makes the CFD dataset, Crack500 dataset, and a customized dataset called CrackSC available on request. In another study [49], a wide variety of road obstacle datasets are available on request. The road anomaly detection study [50] provides multiple datasets; however, login access is needed to download. The research on the Adaboost algorithm for pavement distress detection [51] provides access to the dataset through the journal's website for readers with paper access. The research on thermal image analysis for defect detection [52] provided the dataset upon request.

Furthermore, we would like to inform you that we have thoroughly reviewed the entire systematic review, addressing any grammatical errors and making additional updates to enhance the clarity of language and ideas. These revisions aim to improve the overall quality and readability of the manuscript.

We are grateful for the valuable comments and appreciate the opportunity to improve our work through these constructive feedbacks.

Reviewer 2 Report

  • The paper has been revised and can be accepted.

Author Response

We appreciate and thank you for taking out time to provide the second review. Upon reviewing your comments, we acknowledge your concerns regarding the adequacy of the method description, presentation of results, and the conclusion section in our systematic review. We would like to inform you that we have thoroughly reviewed the entire systematic review, addressing any grammatical errors and making additional updates to enhance the clarity of language and ideas. These revisions aim to improve the overall quality and readability of the manuscript.

We are grateful for the valuable comments and appreciate the opportunity to improve our work through these constructive feedbacks.